# Geochronological insights of middle miocene primates and vertebrate fauna of Ramnagar (J&K, India): Integrating litho- and magnetostratigraphy

Deepak Choudhary[1,2*], Christopher C. Gilbert[3,4,5,6], Christopher J. Campisano[7,8], Daniel J. Peppe[1], Mohammad Arif[9], Sarvendra Pratap Singh[9], Kahsay Tesfay[1], Biren A. Patel[10,11], Wasim Abass Wazir[2], Rohit Kumar[2], Binita Phartiyal[9], Ningthoujam Premjit Singh[12], Kongrailatpam Milankumar Sharma[13], Rajeev Patnaik[2]

**1** Department of Geosciences, Baylor University, Waco, Texas, United States of America, **2** Department of Geology, Panjab University, Chandigarh, India, **3** Department of Anthropology, Hunter College of the City University of New York, New York, New York, United States of America, **4** PhD Program in Anthropology, The Graduate Center of the City University of New York, New York, New York, United States of America, **5** Division of Paleontology, American Museum of Natural History, New York, New York, United States of America, **6** New York Consortium in Evolutionary Primatology, New York, New York, United States of America, **7** Institute of Human Origins, Arizona State University, Tempe, Arizona, United States of America, **8** School of Human Evolution and Social Change, Arizona State University, Tempe, Arizona, United States of America, **9** Palaeomagnetism Lab, Birbal Sahni Institute of Palaeosciences, Lucknow, India, **10** Division of Integrative Anatomical Sciences, Department of Medical Education, Keck School of Medicine, University of Southern California, Los Angeles, California, United States of America, **11** Human and Evolutionary Biology Section, Department of Biological Sciences, University of Southern California, Los Angeles, California, United States of America, **12** Wadia Institute of Himalayan Geology, Dehradun, India, **13** Department of Geology, Central University of South Bihar, Gaya, Bihar, India

☉ These authors contributed equally to this work.
* deepakc015.dc@gmail.com

## Abstract

The Middle Miocene site of Ramnagar (Jammu and Kashmir), India is well-known for its fossil primates including *Sivapithecus indicus, Kapi ramnagarensis,* and *Ramadapis sahnii.* Although always suggested as similar in age to the Chinji Formation of the Potwar Plateau, Pakistan, no precise geochronological age estimates of Ramnagar have yet been obtained. Furthermore, despite some notable previous efforts, the stratigraphic section in the Ramnagar region has not been satisfactorily documented. Therefore, it has long been unclear exactly how the fossil localities around Ramnagar relate to each other or to other Middle Miocene Siwalik localities stratigraphically or chronologically. Here we present new paleomagnetic and revised stratigraphic data for most of the Ramnagar primate-yielding localities. Additionally, we document a near-complete chronology of all the mammal yielding sites belonging to the main ~440 m thick section of the Lower Siwaliks at Ramnagar. We anchor the paleomagnetic results using existing rodent-based biochronology, and further extrapolate ages using sediment accumulation rates to help constrain possible correlations to the Geomagnetic Polarity Time Scale (GPTS). Our preferred interpretation constrains all the primate yielding sites from Ramnagar to between ~13.03 Ma and ~11.59 Ma.

**Data availability statement:** All relevant data are within the manuscript and its Supporting information files.

**Funding:** • D.C. received support from the Leakey Foundation (https://leakeyfoundation.org), grant S202410509. • C.G., C.C., B.P., and R.P. received support from the National Science Foundation (NSF) (https://www.nsf.gov), awards BCS 1945736, BCS 1945618, and BCS 1945743. • R.P. received support from the Ministry of Earth Sciences, Government of India (https://moes.gov.in), grant MoES/P.O.(Geosci)/46/2015, and from the Science and Engineering Research Board (SERB), Government of India (https://www.serb.gov.in), grant HRR/2018/000063. • C.G. received support from the PSC CUNY Faculty Award Program, Hunter College (https://www.rfcuny.org). • B.P. received support from the American Association of Biological Anthropologists (AABA) Professional Development Grant Program (https://physanth.org), the University of Southern California (USC) (https://www.usc.edu), and the Institute of Human Origins, Arizona State University (IHO/ASU) (https://iho.asu.edu). The sponsors or funders had no role in study design, data collection and analysis, decision to publish, or preparation of the manuscript.

**Competing interests:** The authors have declared that no competing interests exist.

Moreover, the first appearance datum (FAD) of the stem hylobatid *Kapi ramnagarensis* and sivaladapid *Ramadapis sahnii* can now be placed between 12.88–13.03 Ma (incorporating two possible correlations). Further, the FAD of the great ape *Sivapithecus* can be pushed back into this same time range as well, thereby extending its known chronological range by up to 200,000 years.

## Introduction

The Middle Miocene Lower Siwalik deposits near the town of Ramnagar (Udhampur District, Jammu and Kashmir, India (Fig 1 a and b) are well known for the presence of a diverse mammalian assemblage that includes at least four primate species: *Sivapithecus indicus*, *Kapi ramnagarensis*, *Ramadapis sahnii*, and *Sivaladapis palaeindicus*. For over a century, the Ramnagar fossils have been biostratigraphically correlated with the Middle Miocene Chinji Formation on the Potwar Plateau of Pakistan [1–20]. The Chinji Formation type locality of the Potwar Plateau is situated ~370 km west of Ramnagar, and it has been dated to ~14.0–11.5Ma using combination of lithostratigraphy, magnetostratigraphy, fission track dating, and biochronology [21–25]. However, despite more than 100 years of fieldwork in the Ramnagar region, the precise geochronologic ages and stratigraphic positions of the Ramnagar fossil localities are unclear due to the absence of detailed lithostratigraphic and geochronological studies comparable to those on the Potwar Plateau. Given the presence of multiple primate taxa, in particular fossil apes representing likely early relatives of both the Asian great ape (i.e., *Sivapithecus*) and lesser ape (i.e., *Kapi*) lineages [17,19], along with the broader mammalian fauna, a more precise understanding of the Ramnagar geochronology is important for documenting the timing of primate and mammalian evolution, including the potential first appearance datums (FADs) of a number of mammalian lineages.

In this study, we combine multiple lines of evidence, including lithostratigraphy, magnetostratigraphy, and biochronology to provide updated possible correlations to the Geomagnetic Polarity Time Scale (GPTS) and to the Chinji Formation on the Potwar Plateau. Our geological fieldwork entailed traversing the entire Lower Siwalik and Middle Siwalik sequence exposed around Ramnagar, and placing key fossil mammal localities within the stratigraphic sequence. We documented the complete ~440 m stratigraphic sequence around Ramnagar and constructed a local magnetic polarity section of 190 m that covers nearly all of the well-known fossil primate and other mammal localities. We then used well-established micromammal biochronology from the Potwar Plateau [23,26–28] to correlate our local polarity stratigraphy to the GPTS [29] and to determine the age and estimate sediment accumulation rate of the entire Ramnagar succession, and the key fossil sites within the sequence. Using these age constraints, we then compared the age ranges of Ramnagar primate taxa to those found on the Potwar Plateau to examine regional patterns of the first and last occurrence.

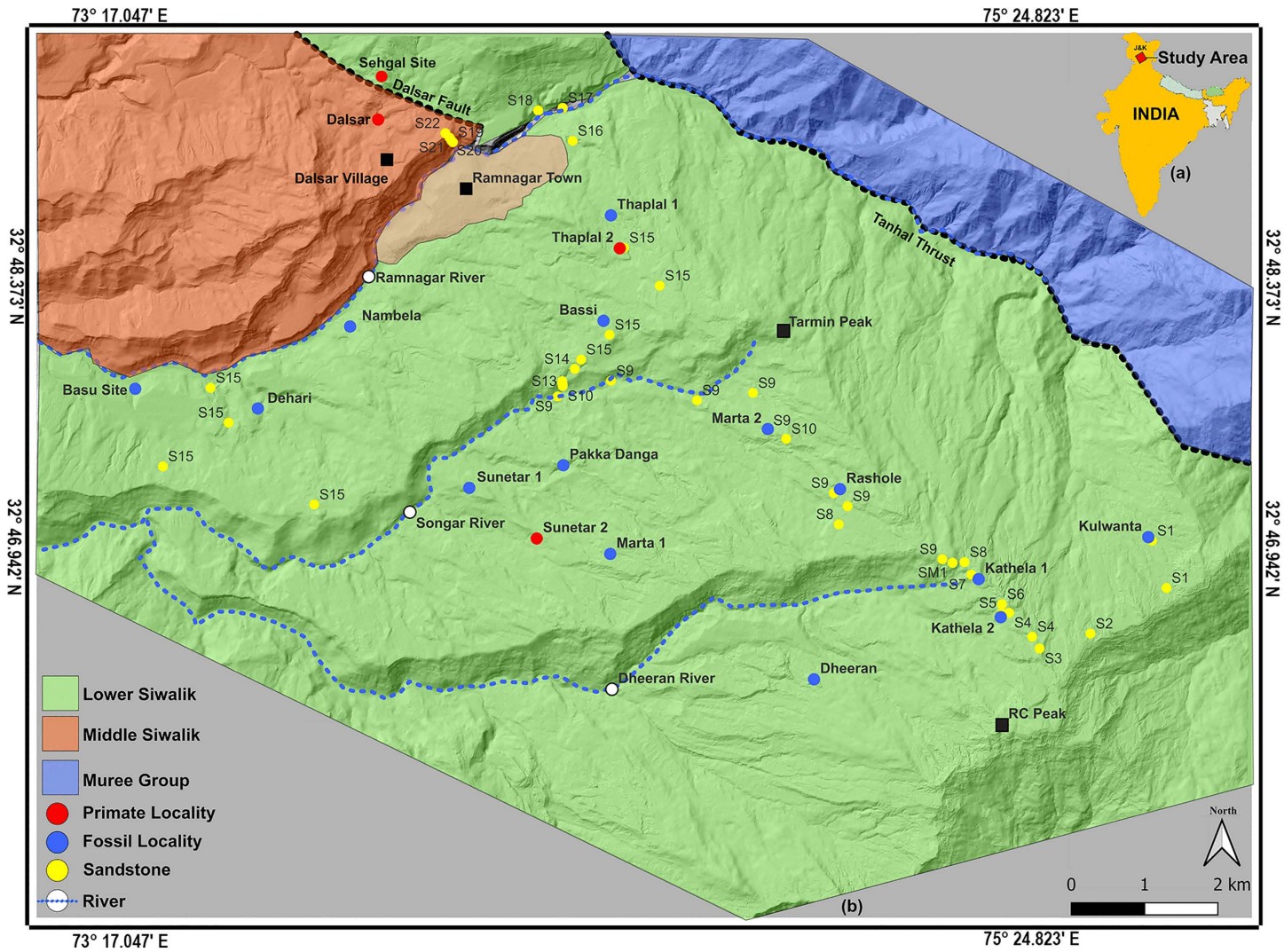

**Fig 1. Geological and geographical setting of the study area.** (a) Geographical position of Ramnagar within Jammu & Kashmir, India. (b) Detailed geological map showing major lithological units and fossil-bearing localities of the Ramnagar area. Note: The base map was generated in QGIS software using Precision3D Digital Terrain Model (DTM) high-resolution imagery (Vantor/Maxar Technologies, Copyright 2026). Geological boundaries are redrawn and modified after Basu (2004).

## Geology of the area

The Siwalik Group represents fluvial deposition from higher altitudes into the Himalayan foreland basins. With an estimated thickness of more than 5,000 meters, the Siwalik sequence in the Jammu and Kashmir region is divided into four mappable lithostratigraphic units: the Mansar Formation, which consists of the Dodenal Member and the Ramnagar Member (Lower Siwalik,), the Dewal and Mohargarh Formations (Middle Siwalik), and the Uttarbaini Formation (Upper Siwalik) [30,31]. The Mansar Formation lies above the Early Miocene Murree Formation. The type section is exposed in the Mansar area, after which it is named [30]. The reference section of the Mansar Formation is exposed in the Ramnagar area between Ramnagar town and Ramchand Peak (RC Peak) (Fig 1b), RC Peak is situated ~11 km southeast of Ramnagar town. The section is continuous, as it lies along the southern limb of Udhampur syncline and the strata gently dip toward

Ramnagar town from RC Peak, with an average dip of 12° northwest and strike direction north-northeast to south-southwest (N42E). The strike length of this section is approximately 12 km, extending from Dehari (West) to Kulwanta Chowk/Junction (East). Due to the orientation of the dip and the pattern of erosion in the area, the oldest sediments are exposed at the highest elevation (Ramchand Peak), with the youngest exposed at the lowest elevation (Ramnagar Khad/River). The sequence then continues across the river, ascending topographically to Dalsar and transitioning into the Middle Siwaliks as defined lithologically by multi-storied sandstone. The Ramnagar member of Mansar formation is composed of alternating beds of arenaceous (sandstones, intraformational sandstones, and conglomerates) and argillaceous (clay and siltstone paleosols) rocks. A previous study [7] documented a series of 10 sandstones in a ~350 m sequence, however, fieldwork conducted by our group identified several discrepancies between the published work and outcrop stratigraphy, which this study aims to resolve.

## Methodology

### Lithostratigraphy

Since 2020, our team has conducted multiple field seasons at Ramnagar, focusing particularly on lithostratigraphy, magnetostratigraphy, and paleontological collections. We undertook multiple traverses following marker beds from Kulwanta to Ramchand Peak, Ramchand Peak to Rashole, Ramchand Peak to Dheeran, Rashole to Sunetar-1, Sunetar-1 to Bassi, Bassi to Dehari, Bassi to Thaplal, Thaplal to Ramnagar Khad, and Ramnagar Khad to Dalsar to determine the complete lithostratigraphic thickness of the section, to describe the Mansar Formation lithofacies, and to corelate between key fossil sites. During these traverses, we walked out bedding plane contacts and marker beds, which were primarily large sandstone units, between the sections. These efforts resulted in a 440 m stratigraphic sequence from Ramchand Peak to Dalsar. The geological section was measured using a combination of a Jacob's staff with an Abney hand-level on traversable sections and a TruPulse L2 laser rangefinder and a 10-m measuring tape at road cuts and outcrop exposures that were too steep to climb. The dip and strike of each sandstone were determined using a Brunton Pocket Transit Compass. A Garmin handheld GPS unit was used to record the latitude, longitude, and elevation of each locality and sandstone.

Twenty-two large intraformational sandstones were identified and numbered in stratigraphic order from lowest to highest across the section (S1-S22) (see Figs 1.b). Where possible, we tried to synonymize the sandstones recognized here with those previously identified and published [7] (Table 1). From the complete 440 m stratigraphic column, we sampled a 190 m thick section exposed along the Songar River and Ramnagar River for magnetostratigraphy that spans many recently described fossil localities, and also sampled section at three primate-bearing localities: Sunetar 2, Rashole, and Thaplal [7,8,10,13,16–20].

### Magnetostratigraphy and magnetic mineralogy

Paleomagnetic sampling was conducted through a composite 190-m thick section encompassing all known primate and major fossil localities in the Middle Miocene Ramnagar Member. This includes the Main Section (MS), which extends from S9 to S15, and the Dehari Section (DH), which extends from S15 to S16 (Fig 1.b). As the ~35 m paleosol sequence between S15 and S16 is more continuous and better exposed at the Dehari section than MS, it was preferred for paleomagnetic sampling. Samples were also collected from Thaplal (TH), Sunetar-2 (ST), Rashole (RS), and Bassi (BS) for comparative analysis and to assess lithostratigraphic correlation between fossil localities. Paleomagnetic samples were collected from fine-grained mudstone (paleosols/mudstones) and sandstones at intervals ranging from 0.25 to 8 m, with an average interval of 1–2 m. In total, 241 block samples were collected from 84 sites: 172 samples from the Main Section (55 sites), 37 from Dehari (16 sites), 11 from Sunetar-2 (6 sites), 12 from Thaplal (4 sites), 5 from Rashole (2 sites), and 4 from a single site at Bassi (details in S1 Table). Multiple bedding orientations (strike and dip) were recorded for tilt corrections from each sampled stratigraphic section. The strata gently dip northwest at an average of 12° from Ramchand Peak toward Ramnagar town, with a strike of N42°E.

**Table 1. Association of numbered sandstones from this study (S1 = oldest) with fossil localities, paleosol horizon and primate taxa. Where possible, correlations between sandstones of Basu [7] are noted. Note that if Dalsar represents the Middle Siwaliks, the Sivapithecus specimens recovered could represent S. sivalensis.**

| New Section (This Study) | Basu Section | Localities Above these sandstones | Fossiliferous Paleosol Horizon | Primate Taxa |
|---|---|---|---|---|
| S22 | | Dalsar | | *Sivapithecus* sp. |
| S17 | A | – | | |
| S16 | B | – | | |
| S15 | E & I | Thaplal & Dehari | P18 | *Sivapithecus indicus, Sivaladapis palaeindicus* |
| S14 | – | Bassi | P17 | |
| S12 | G | Tarmin Peak | P15 | |
| S9 | – | Sunetar, Rashole, Marta | P9 | *Sivapithecus indicus, Kapi ramnagarensis, Ramadapis sahnii* |
| S6 | – | Kathela 2 | P6 | |
| S5 | – | Kathela 1 | P5 | |
| S4 | – | Ritti Mitti 2 | P4 | |
| S2 | | Ritti Mitti 1 | P2 | |
| S1 | – | Kulwanta | P1 | |

Samples were collected from both fine-grained sediments (clays, siltstones and paleosols) and sandstones in the section. Well-indurated sandstone samples were collected by measuring the strike and dip of the bedding surface with a Bruton Pocket Transit Compass and then removing a large block. At these sample sites, one large block sample (up to ~0.5 m in diameter) was collected. Clays, siltstones, paleosols, and fine-grained sandstones were also sampled at higher resolution between block sampled sandstone units. Prior to sampling, the finer grained material, sampling sites were trenched to remove surface weathering so that unweathered material could be sampled. For the fine-grained units, when sampling an *in situ* flat surface was prepared using a hand rasp. The strike and dip of the prepared face was measured with a Brunton Pocket Transit. At each sampling horizon ("site"), typically four block samples ("specimens") were collected.

Samples were prepared in two ways. The sandstone block samples were subsampled by core drilling and sectioning into standard cylindrical specimens (2.54 cm diameter, 2.2 cm height). Up to six cores per sample were sub-sampled from the sandstone block samples (21.5% of samples with prefix D21 see S2 Table). For the finer grained samples and resampled sandstone (79.5% of samples with prefix D23 and D25), at least three of the four collected sample specimens were cut into 2–4 cm³ cubes using a lapidary saw, and each sample produced a single specimen for analysis.

Paleomagnetic measurements and natural remanent magnetization (NRM) of the finer grained samples were conducted using a 2G Enterprises cryogenic DC-SQUID magnetometer inside a two-layered magnetostatic shielded room with a background field typically less than 300 nT at Baylor University, Texas, USA (BU). For the sandstone block samples, measurements were performed using a JR6 spinner magnetometer at the Birbal Sahni Institute of Palaeosciences, Lucknow, India (BSIP). Samples were demagnetized following a similar protocol at BU and BSIP. The NRM of each sample was measured before step-wise thermal demagnetization. All specimens were stepwise demagnetized using an ASC Scientific TD-48 thermal (Th) demagnetizer with increment intervals of 25–50°C up to a maximum unblocking temperature, or until magnetization became erratic and unstable, which was typically between 650 and 680°C. Progressive demagnetization continued until the magnetic intensity dropped below the noise level or the directions became unstable. To minimize oxidation reactions in the fine-grained samples, thermal demagnetization was conducted in a controlled nitrogen atmosphere.

The demagnetization data from BU were analyzed using the PMGSC software [32–34] and the data from BSIP were analyzed using an AGICO Remasoft 3.0 software [35]. The characteristic remanence (ChRM) of each specimen was calculated using the orthogonal endpoint Zijderveld diagrams technique [36] and the least-squares best fitting line following

the principal component analysis (PCA) approach [37] or remagnetization of great circles analysis [38,39]. The ChRM was extracted from at least four remanent vector endpoints aligned linearly toward the origin and had a maximum angle of deviation (MAD) < 20°. Demagnetization data were typically anchored to the origin because most specimens exhibited a well-defined stable endpoint demagnetization behavior that showed a trend directed to the origin. Specimens with ChRM directions with MAD > 20° and specimens with erratic demagnetization behavior were excluded from the further analysis and not used in our polarity determinations. Samples with erratic or poorly defined directions accounted for less than 6% of analyzed samples. The site mean direction for sample horizons with three or more significant ChRM directions was calculated using Fisher statistics [40] and the site mean ChRM directions were converted to virtual geomagnetic pole (VGP) latitudes. Site mean direction with an $\alpha_{95} > 35°$ did not pass the Watson test [41] for randomness and were excluded from further interpretation. Reversal boundaries were placed at the stratigraphic midpoint of samples of opposing polarity. The resulting local magnetic polarity stratigraphy (LMPS) was then correlated to the geomagnetic polarity timescale (GPTS) [29].

A triaxial isothermal remanent magnetization (IRM) Lowrie test [42] was conducted on 11 samples collected from different lithologies to identify the mineralogy of the primary magnetic carriers. A 1 T, 300 mT, and 100 mT field was imparted along the X, Y, and Z axes using an ASC Scientific impulse magnetizer. Samples were then thermally demagnetized in 20–100°C incremental heating steps up to 700°C at BU using an ASC Scientific controlled atmosphere thermal demagnetizer in an $N_2$ atmosphere. The magnetization of the X, Y, and Z axes was then measured using a 2G Cryogenics DC-SQUID magnetometer at each demagnetization step.

The individual(s) pictured in the Supporting Information figures have provided written informed consent (as outlined in PLOS consent form) to publish their images alongside the manuscript.

## Results

### Lithostratigraphy

The complete lithostratigraphic column of the Ramnagar sequence from Kulwanta [10,15,18] to Dalsar localities [9] consists of 22 reference sandstones (numbered S1 to S22 from the bottom to top), 26 reference paleosol/mudstones (P1 to P26), as well as multiple silty mudstones, intraformational sandstones, and muddy mottled sandstones (Fig 2, S1-S9 Fig). This is in contrast to the 10 sandstones (identified as A-J) documented by Basu [7]. In some cases, sandstones recognized as distinct by Basu [7] were demonstrated to be the same sandstone based on physical tracing of the units, while other recognized here were undocumented by Basu [7] (see Table 1), particularly at higher elevations (e.g., above Tarmin Peak). Additionally, some units recognized by Basu [7] either were not in our stratigraphic succession or were unable to be located using published maps (e.g., Sandstones C and J in Fig 2, [7]).

The 22 reference sandstones in our study range from 1.5 to 13 m thick and are greyish and dark brown to yellowish in color with a medium to fine grain size (S1 Fig). The brownish sandstones are typically coarser grained than the greyish sandstones. The other sandstone beds, which we refer to as intraformational sandstone units are thinner (typically <2 m thick) fine to very fine-grained sandstones that are grey to greyish pink in color. These intraformational sandstones cannot be traced laterally over long distances. The sandstone bodies within our section can be categorized into three distinct types based on their thickness and lateral continuity, reflecting diverse fluvial depositional environments. Type 1 sandstones are thick (5–6 m or more), laterally extensive sandstone units traceable over kilometers, interpreted as major channel-belt deposits of a meandering or low sinuosity river system. Type 2 sandstones include moderately thick sandstones (2–6 m) with limited lateral extent (hundreds of meters), representing crevasse splays or minor distributary channels formed during episodic flood events. Type 3 sandstones consist of thin (<2 m), laterally restricted sandstone lenses, likely deposited as distal splays, sheetfloods, or abandoned channel fills in low-energy floodplain settings. Sandstone types 1 and 2 comprise our 22 reference sandstones, while type 3 sandstones are the intraformational sandstones. In particular, three sandstone units (S1, S9, S15) serve as key stratigraphic markers due to their erosion-resistant nature,

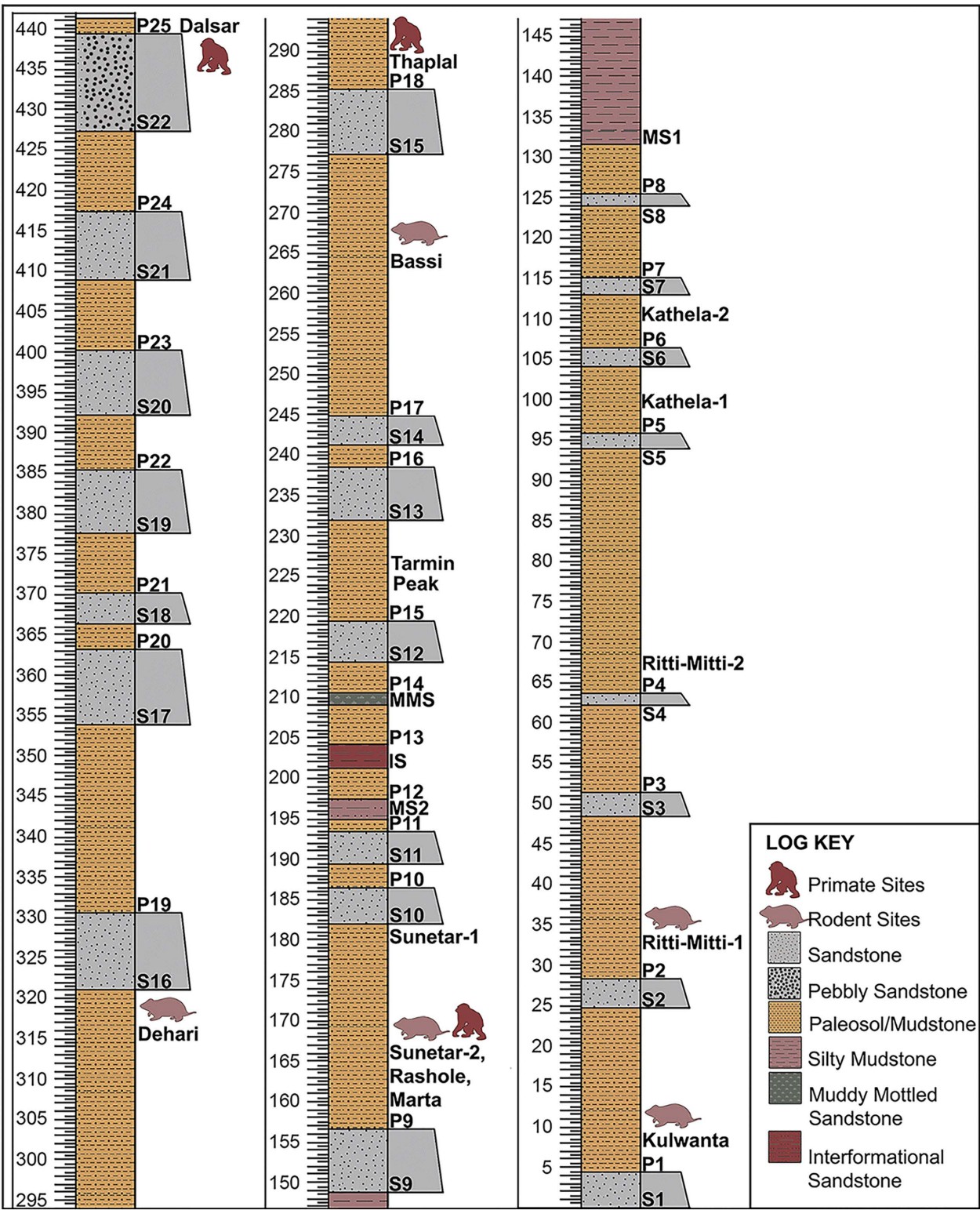

**Fig 2. Complete lithostratigraphic column of the Ramnagar section, illustrating major fossil-bearing localities.** Paleomagnetic sampling was conducted across a 190 m interval of the sequence, spanning units S9 to S16. Sandstone unit S1 represents the oldest horizon in the section, while S22 is the youngest. S22, located below Dalsar locality, belongs to the Middle Siwalik formation (see Fig 1c for reference).

thickness and lateral continuity (traceable over kilometers). These units act as basement of prominent cuesta-type landforms, with the S1–S9 sandstone sequence creating escarpments along the Dheeran River, the S9–S15 sequence exposed along the Songar River, and the S15–S22 sequence traceable at Ramnagar Khad/River.

The majority of the fine-grained deposits are pedogenically modified and the lithology was claystone and siltstone, and occasionally sandy siltstone or very fine silty sandstone. These fine-grained units, which we colloquially refer to as paleosols because the majority have at least some degree of pedogenic modification, occur between the sandstone units and range from 5 to 30 m in thickness.

Overall, the Ramnagar sequence is dominated by fine overbank deposits that are variably pedogenically modified and interbedded with interval of thick, laterally continuous sandstones that likely represent major fluvial channel or sheet flood deposits that covered the landscape (Fig 2, S5 Fig). At the base of the lithostratigraphic column lies Sandstone 1 (S1), which appears to conformably overlie the Murree Formation. Above this sandstone lies the Kulwanta fossil locality, which is situated on the northwest side of the road from Kulwanta Chowk to Ramchand Peak (RC peak Fig 1). The youngest sandstone, S22, lies on top of cuesta-type topography along the western side of the Ramnagar Khad, with the Dalsar village lying atop this sandstone. This uppermost pebbly sandstone (S22) displays characteristics that distinguish it from other sandstones in the section (S2-S3 Figs). S22 consists of massive, multistoried blue-grey sand with reduced red silt content and a high concentration of quartz pebbles. Lithologically, it aligns more closely with the sandstones found in the Middle Siwaliks rather than with those of the Lower Siwaliks, which are generally thinner, devoid of quartz pebbles, and lack evidence of multistoried channel structures (e.g., [23,43]).

## Magnetic mineralogy

Multiple-component IRM-Th demagnetization experiments [42] (Fig 3a and 3b) for all analyzed samples indicate that samples were dominated by a similar mineralogy through the Ramnagar section. The vast majority of IRM was held by grains with a coercivity of 1T (Fig 3). In all analyzed samples, the remanent magnetization of the hard coercivity component (1

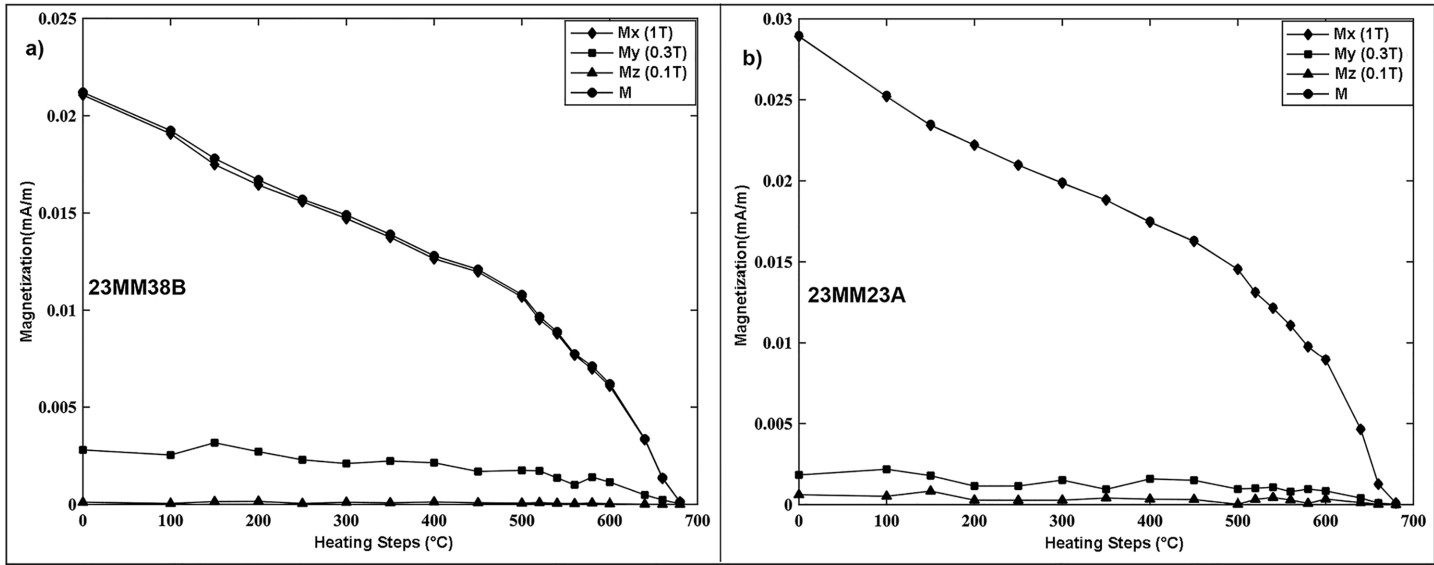

**Fig 3. Rock magnetic properties of selected specimens (a) and (b) are the progressive thermal demagnetizations of a composite isothermal remanent magnetization (IRM) [42] that were produced by magnetizing the specimens in a magnetic field Mx, My, and Mz of 1T (high coercivity or hard component), 0.3T (medium coercivity component), and 0.1T (low coercivity or soft component), along three mutually orthogonal axes, respectively.**

T) decreased sharply between 650°C and 700°C indicating the presence of hematite (Fig 3). Whereas the soft (0.1 T) and intermediate coercivity components (0.3 T) show a more gradual decline, with unblocking temperatures between 200°C–350°C and remanence occasionally persisting above 400°C, and complete demagnetization between ~650–680°C suggesting potential contributions from greigite and magnetite (Fig 3). In addition to the IRM-Th demagnetization behavior, most demagnetization temperatures align with Curie values ≥ 675°C, also indicating that hematite is the primary magnetic carrier in the sediments (Fig 4).

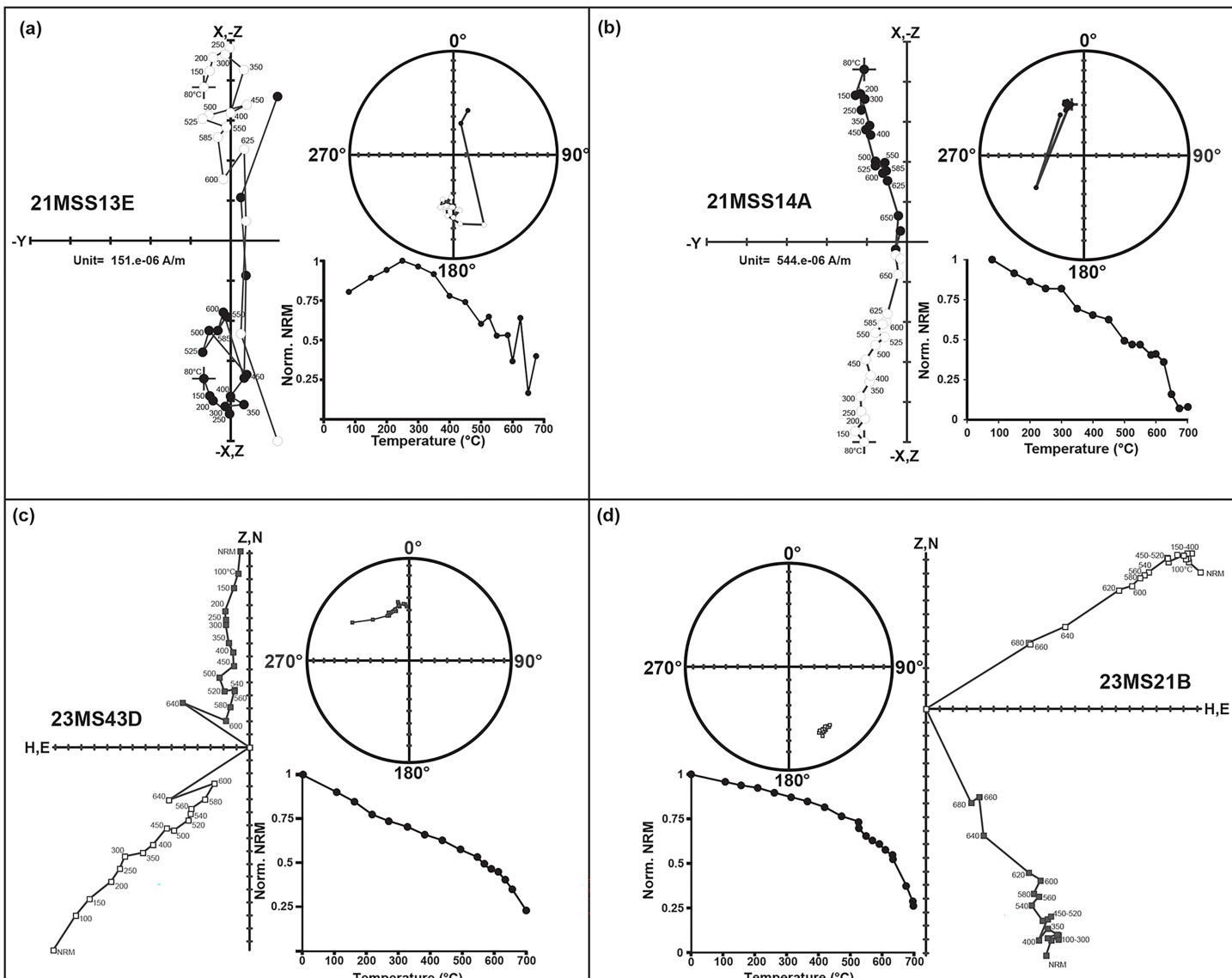

**Fig 4. Representative Zijderveld diagrams, NRM intensity decay, and equal-area plots showing thermal demagnetization after bedding tilt correction.** (a, b) Reversed and normal polarity trajectories for sandstones. (c, d) Reversed and normal polarity trajectories for paleosols. Solid symbols: downward inclinations; open symbols: upward inclinations. Magnetization intensity scales: × $10^{-2}$ A/m.

## Magnetostratigraphy

Two hundred and fourty One block samples were collected from eighty-four84sites and analyzed during this study (S1 Table). The typical demagnetization results obtained are shown in Fig 4, indicating that the characteristics of high-blocking temperature magnetization directions are adequately defined during thermal demagnetization, yielding populations of normal and reversed magnetization components before and after bedding tilt correction (Figs 4 and 5). The Th demagnetization curves of most specimens typically exhibit two-component behavior: a low-temperature remanent component (LTC) in almost all the samples that was removed by approximately 350°C, and a high-temperature component (HTC) that was isolated between 350 and 675°C. Although the majority of samples had a single directional component that lacked a clear separation between the LTC an HTC (Fig 4 b and 4c)

The stepwise Th demagnetization enabled isolation of the ChRM component for the majority of samples analyzed (98.3%), which allowed construction of a magnetic polarity stratigraphy (Fig 5). A small number of samples (4%) exhibited erratic demagnetization behavior above 500°C, preventing the isolation of reliable ChRM directions. The HTC direction, including antipodal normal and reversed polarities, shows a well-defined stable linear segment that is directed toward the origin and is interpreted as characteristic remanence direction (ChRM) (Figs 4 and 5). A total of 98.3% (241 samples) of reliable ChRM directions were determined as PCA-derived linear directions (Fig 5a, S2 Table). In some cases (1.7%), due to the overlapping LTC and HTC, a Great Circle (GC) fit was used to determine a ChRM vector (S2 Table). Site mean directions generally showed good within-site consistency (Fig 5b, S3 Table). Despite some scattered sites, we calculated the site mean for fourty-seven sampling horizons (57.3% of sampled horizons) that passed Watson's (1956) randomness test and these sites mean directions were used for our polarity interpretations (S3 Table). The sites mean and specimen directions establish a local polarity stratigraphy for the Ramnagar sequence (Fig 6). The average specimen and site mean direction distributions are clustered into two opposite polarity magnetization directions for each magnetic polarity chron from Main Section, Dehari and Thaplal l (Fig 5c) (Table 2). A reversal test [39] indicates that the site-mean directions do not pass at the 95% confidence level. The deviation from a perfect dipole in the site-mean directions is attributed to the structural tilt of the southern limb of the Udhampur Syncline, where bedding dips 12–15° and the apparent dip ranges between 3–5°. Such local deformation alters the orientation of the tilt corrected magnetization directions, likely causing the observed directions to diverge from those expected for an ideal geocentric axial dipole.

The 190 m main section includes five normal polarity events separated by four reversals, with lithostratigraphy, local polarity stratigraphy, specimen/site-mean polarity, and VGP latitudes summarized in Fig 6. All samples from Dehari, Thaplal, Rashole, and Sunetar are reversed. These sections were correlated to the main Ramnagar section based on lithostratigraphic correlation to develop a composite magnetostratigraphic section for the complete Ramnagar section (Fig 6).

## Discussion

### Relationship of polarity stratigraphy to GPTS

The local polarity stratigraphy demonstrates nine polarity intervals through the Ramnagar section (Fig 6). Most of the reversals have multiple site mean directions (R1, N2, R2, R3, N4, R4, N5; Fig 6). However, two intervals are defined by a single site mean (N1, N3) and both of these normal intervals correspond to major sandstone units (Table 2) (Fig 6). It is possible that these two normal intervals (N1 and N3) could sample overprint directions; however, we deem this highly unlikely for four reasons. First, we only sampled fine grained sandstone beds and there are multiple sandstone units within the stratigraphy that have the same polarity as the surrounding samples from fine-grained paleosol units, including both normal (S15, S16) and reversed direction (S10, S11, S13) (Fig 6). Second, the magnetic mineralogy of the sandstone and fine-grained paleosol units are identical (Figs 3 and 4), demonstrating that they have the same magnetic carrier making it unlikely that the sandstones were overprinted while the fine-grained paleosols were not. Third, samples from different horizons within the sandstone units without site mean directions show a direction that is

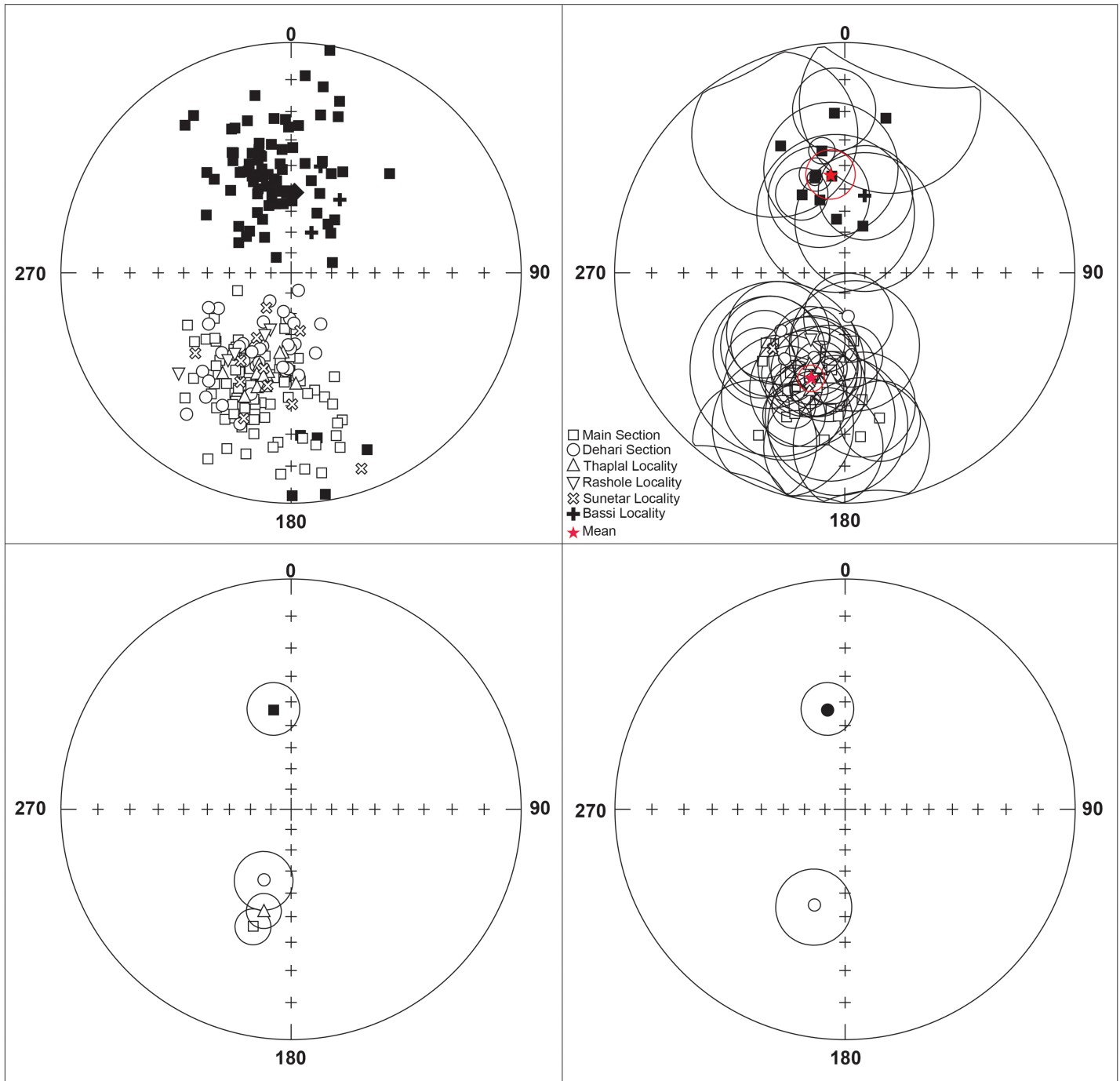

**Fig 5. The Ramnagar section's characteristic remanence (ChRM) directions.** (a) Bedding-tilt corrected ChRMs of the specimens that roughly converged into two antipodal clusters. (b) Site mean direction for normal and reverse polarity. (c) Normal and reversed site mean directions from Main Section, Dehari Section, Thaplal Locality. (d) Overall mean direction for both reversed and normal polarity. All Solid symbols indicate downward (positive) inclinations, while open symbols denote upward (negative) inclinations.

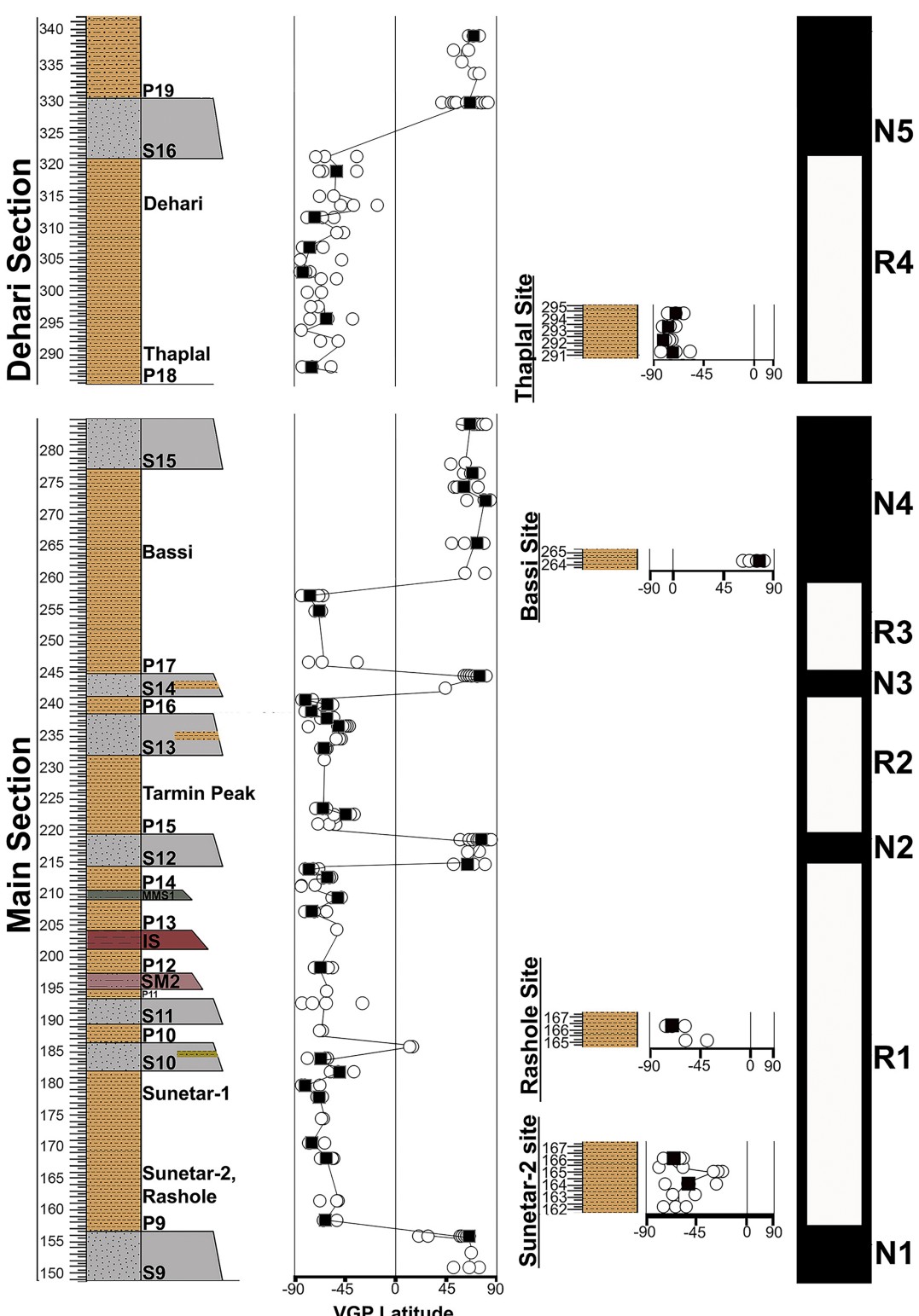

**Fig 6. Virtual Geomagnetic Pole (VGP) Latitude of S9–S16 Sandstones of Main Section and Individual Sites with Correlation to Local Polarity Stratigraphy.**

**Table 2. Mean Paleomagnetic Direction Data.**

| Subset | n | D (°) | I (°) | K (°) | a95 (°) | Pole (N°) | Pole (E°) |
|---|---|---|---|---|---|---|---|
| Sample location stratigraphy | | | | | | | |
| Main section – Reversed | 22 | 198 | −33.8 | 20.9 | 6.9 | −68.4 | 022.4 |
| Main Section – Normal | 13 | 350 | 42.7 | 18.3 | 11 | 78.1 | 305.4 |
| Dehari Section – Reversed | 6 | 201 | −53.7 | 27 | 13.1 | −72.5 | 334.8 |
| Thaplal Locality – Reversed | 4 | 195 | −41.1 | 163 | 7.2 | −73.9 | 016.4 |
| Bassi Locality – Normal | 1 | 14.5 | 51.8 | | | 77.8 | 163 |
| Rashole Locality – Reversed | 1 | 206.8 | −53.8 | | | −67.7 | 333.9 |
| Sunter-2 – Reversed | 2 | 210 | −39.7 | | | −61.6 | 358.8 |
| Local polarity stratigraphy | | | | | | | |
| Normal 1 (N1) | 1 | 341.2 | 53 | | | 74.3 | 353.3 |
| Reversal 1 (R1) | 15 | 195.1 | −34.4 | 18 | 9.6 | −70.6 | 027.4 |
| Normal 2 (N2) | 2 | 343.6 | 42.2 | | | 73.4 | 319.2 |
| Reversal 2 (R2) | 8 | 202.8 | −32.5 | 28 | 10.7 | −64.5 | 016.2 |
| Normal 1 (N3) | 1 | 342.5 | 43.5 | | | 73 | 324 |
| Reversal (R3) | 2 | 188.6 | −34.9 | | | 74.4 | 043.6 |
| Normal 4 (N4) | 8 | 353.1 | 39.5 | 16.33 | 14.1 | 78 | 287.5 |
| Reversal 4 (R4) | 10 | 198.2 | −48.6 | 33.7 | 8.4 | −74.1 | 352.2 |
| Normal 5 (N5) | 2 | 331.4 | 47.8 | | | 65 | 343.9 |
| All data | | | | | | | |
| All normal sites | 14 | 351.8 | 43.7 | 18.5 | 10.4 | | |
| All reversed sites | 35 | 197.6 | −38.7 | 21.4 | 5.4 | | |

(Note: n—number of site means; D—declination; I—inclination; k—Fisher's (1953) precision parameter; a95—radius of 95% confidence cone around mean [40]; pole °N and °E—mean virtual geomagnetic pole latitude and longitude calculated from each subset).

consistent with the site mean. Fourth, the direction data is very robust as the site mean directions are defined using ≥5 specimen directions, and the site mean direction overlaps with the expected Miocene pole direction [44]. Thus, we interpret the reversal sequence, including the two normal intervals with a single site mean direction, to be robust and to reflect Miocene polarity.

While there are no radiometric dates from the section, the local magnetic polarity stratigraphy can be correlated with the GPTS [29] using rodent and macromammal biochronology to constrain the depositional age of the sequence (Fig 7 and 8).

From the bottom of the magnetostratigraphic section, Rashole records the thryonomyid *Paraulacodus indicus* from above S9 [45] (S10 Fig), which has a restricted range on the Potwar Plateau between ~13.1–12.7 Ma [27]. At Tarmin Peak, which is stratigraphically above S12 (or Sandstone G of Basu 2004), reports of *Antemus chinjiensis* (~13.8–12.7) and *Megacricetodon* cf. *sivalensis* (~16.8–12.7) by Sehgal and Patnaik [9] provide an estimated age between ~13.8–12.7 Ma [27,46]. However, additional specimens, particularly more diagnostic first molars, are needed to confirm the presence of *Antemus* at Tarmin Peak relative to the younger and more variable sample of post-*Antemus* beginning ~12.4 Ma [28] (J. Flynn, personal communication). Furthermore, there is a gap in fossil rodent localities between ~12.7–12.4 Ma on the Potwar Plateau [28], making it possible that the true LAD of *Antemus* lies somewhere in this interval, postdating the 12.7 Ma last occurrence at Potwar. Post-*Antemus* samples are first documented at Potwar sites ~12.4 Ma. Nevertheless, given an established age range of ~17.9–12.7 Ma for the genus *Megacricetodon* on the Potwar Plateau, the presence of *Megacricetodon* at Tarmin Peak is suggestive that the site is indeed ~12.7 Ma or greater, or at least greater than ~12.4 Ma when post-*Antemus* occurs and *Megacricetodon* is definitively no longer sampled.

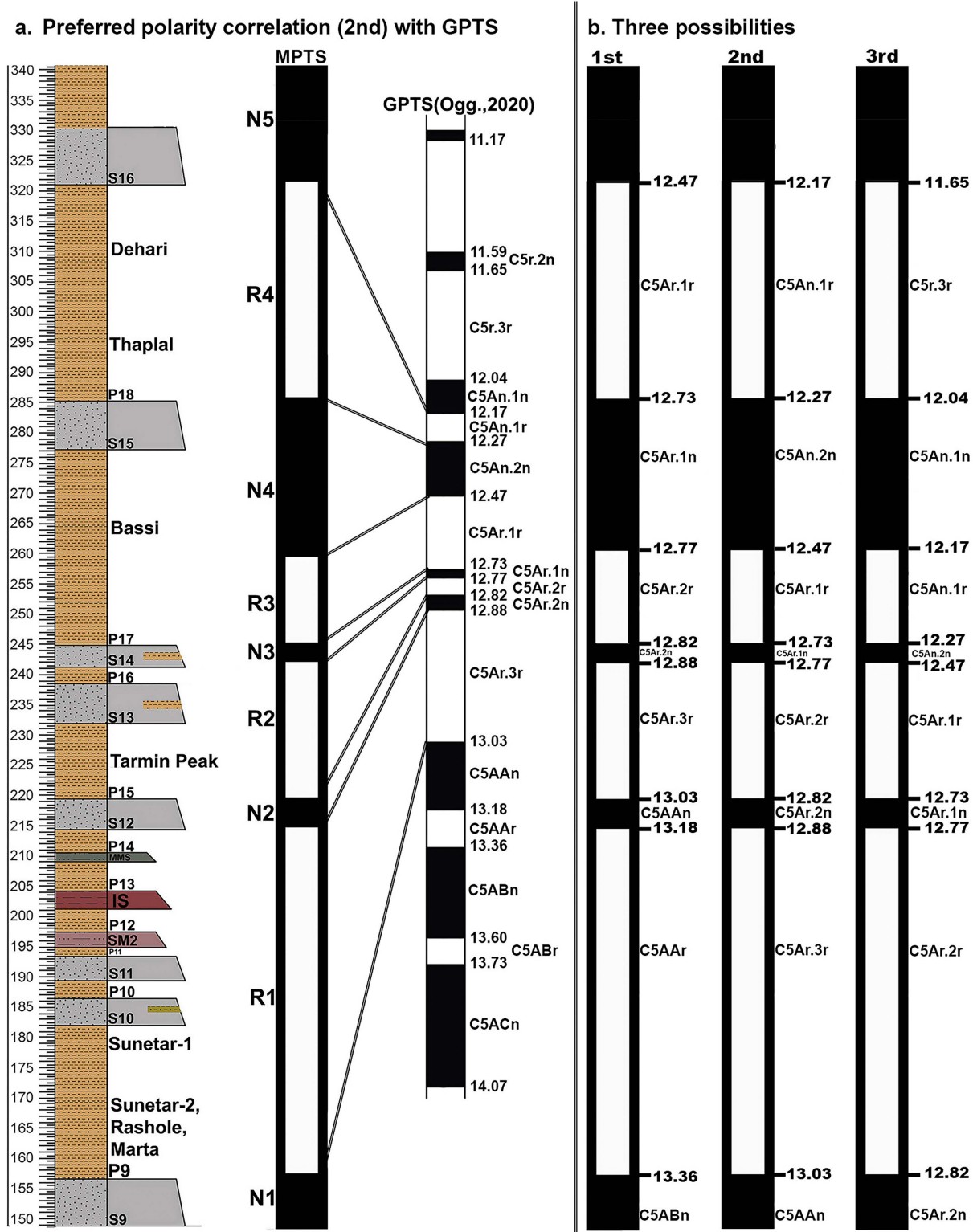

**Fig 7. a) Magnetic polarity stratigraphy of sandstone units S9 to S16, showing the preferred correlation (2nd) with the Geomagnetic Polarity Time Scale (GPTS) [29]. (b)** Three alternative GPTS interpretations proposed and assessed using biostratigraphic constraints from the Ramnagar section.

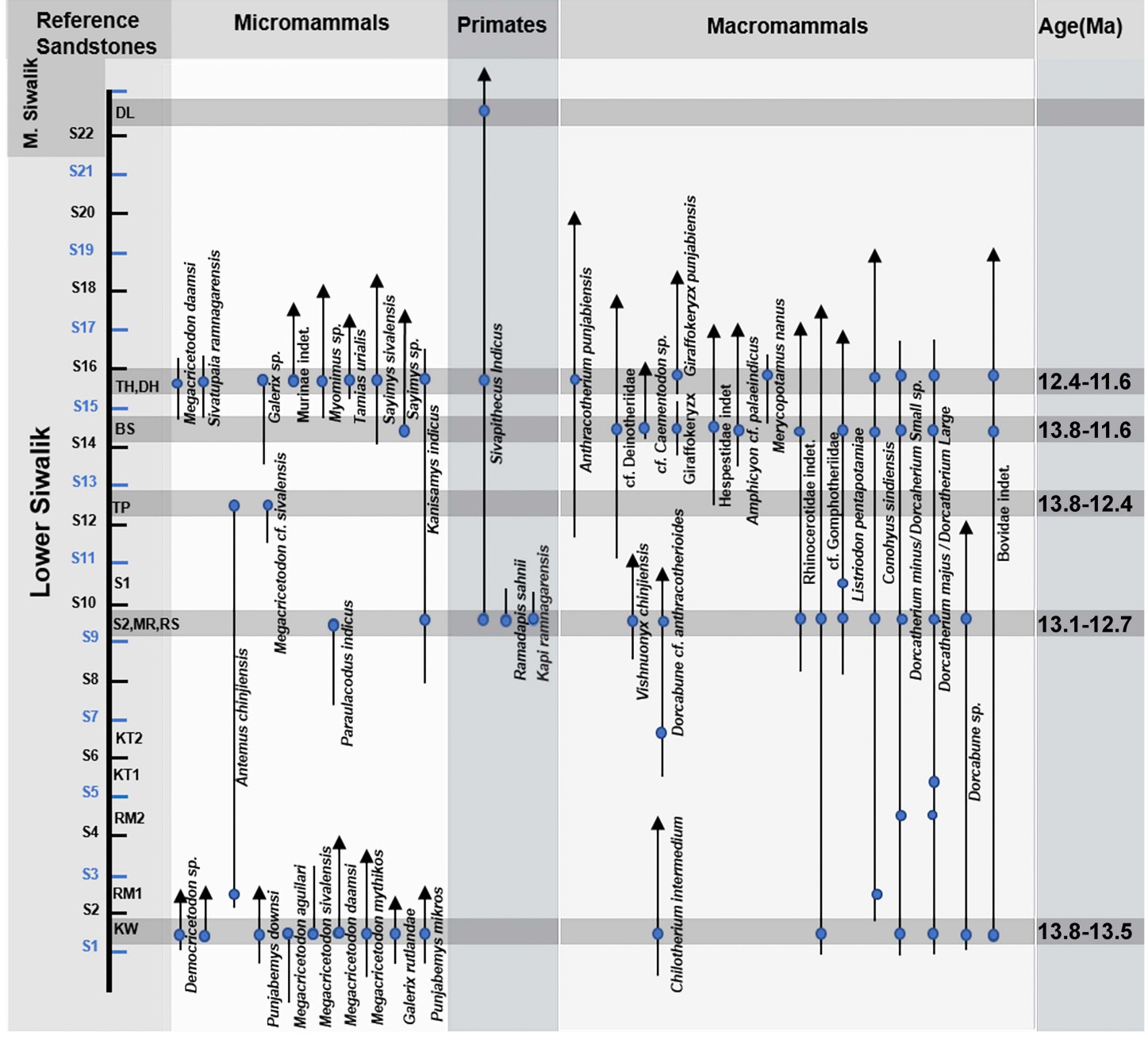

**Fig 8. Biostratigraphy chart of Ramnagar fauna.** DL- Dalsar, TH- Thaplal, DH- Dheari, BS- Bassi, TP- Tarmin Peak, S1- Sunetar 1, S2- Sunetar 2, MR- Marta, RS- Rashole, KT1 – Kathela 1, KT2- Kathela 2, RM- Ritti-Mitti, KW- Kulwanta. Confirmed faunal occurrences in the stratigraphic context based on [8–20,45].

At the site of Bassi, located within the thick paleosol P17 overlying S14, specimens of *Antemus chinjiensis* have recently been described from the lower level [14], suggesting a biochronological age ~ 13.8–12.7 Ma [27]. However, the presence of *A. chinjiensis* at Bassi is again based on a small sample size of specimens: a partial lower right $M_2$ reported by Gilbert et al. [18], and a left upper first molar, a right lower first molar, a right lower second molar, and a left lower third

molar reported by Parmar et al. [14]. Given the variability of post-*Antemus* samples (i.e., some specimens look just like *Antemus* while others are more derived), it is prudent to be cautious about identifying *A. chinjiensis* vs. post-*Antemus* in the absence of larger samples, particularly of first molars. Similar to the situation at Tarmin Peak, a larger sample size is necessary to confidently differentiate *Antemus* from post-*Antemus* (L. Flynn, personal communication), and thus the current sample from Bassi is biochronologically consistent with an age range spanning both *Antemus* and post-*Antemu* (~13.8–11.6 Ma) taxa until the appearance of definitive pre-*Progonomys* samples at site from at sites from ~11.6–11.4 Ma [27,46].

At Dehari (above S15), specimens attributed to the "post-*Antemus*/pre-*Progonomys*" (12.4–11.6 Ma) interval have been reported alongside *Megacricetodon daamsi* (15.2–12.8 Ma), *Galerix* sp. (~14.3–11.6 Ma) and *Kanisamys indicus* (16.8–11.4 Ma) [13,16,47]. However, the upper $M^1$ of the specimen referred to *M. daamsi* displays notable morphological and metric deviations from other *Megacricetodon* samples. The tooth is disproportionately large for *Megacricetodon* and instead falls within the metric range typical of *Democricetodon* (see S4 Table). Additionally, the anterocone on the upper first molar ($M^1$) is not divided into two parts, suggesting it is single-lobed rather than bilobed (L. Flynn, personal communication). Thus, it seems more likely that this specimen represents a species within the genus *Democricetodon*, which ranges between ~21–8.7 Ma [27,46], thereby removing any biochronological constraints imposed by *M. daamsi* at Dehari. Based on the presence of post-*Antemus*/pre-*Progonomys* murines, *Galerix* sp., and *Kanisamys indicus*, the age range of the Dehari locality is estimated to be between ~12.4 to 11.6 Ma.

Given these biostratigraphic constraints between 13.1 and 11.6 Ma for the Rashole, Tarmin Peak, Bassi and Dehari sites from oldest to youngest, and because of the numerous paleomagnetic transitions within the Middle Miocene, there are three potential correlations to the GPTS for the Ramnagar section (Fig 7). We describe each scenario from older to younger in detail below, highlighting major reversals and their association with faunal age ranges.

**Option 1.** In this scenario, R1 is correlated with C5AAr (13.18 to 13.36 Ma), R2 with C5Ar.3r (12.887 to 13.032 Ma), N4 with C5Ar.1n (12.735 to 12.770 Ma), and R4 with C5Ar.1r (12.474 to 12.735 Ma). The Rashole locality, situated above S9, is constrained by a biostratigraphic age estimate of approximately 13.1 to 12.7 Ma based on the *Paraulacodus indicus* age range. Thus, Rashole's position near the lower end of R1 implies a first appearance datum (FAD) nearly 200−250 kyr earlier than previously recognized. While not impossible, this GPTS correlation is less consistent with the biostratigraphy at Rashole compared to Options 2 and 3 (see below). More critically, it would necessitate extending the FAD of *Sivapithecus* (reported from Rashole, see [20] by approximately 400−600 kyr, given the recorded FAD for *Sivapithecus* specimens on the Potwar Plateau is found in C5Ar.1n ~ 12.73–12.77 Ma [22,23,48–50]. This correlation also places Rashole and Sunetar 2 within C5Ar.3r, making specimens of the primates *Sivapithecus*, *Kapi*, and *Ramadapis* from these sites between 13.18–13.36 Ma.

In Option 1, both Tarmin Peak (above S12) and the Bassi locality (above S14) are correlated with R2 and N4, which align well with biostratigraphic age estimates if *Antemus chinjiensis* and *Megacricetodon* are indeed present at Tarmin Peak (Chron C5Ar.3r, ~ 12.88–13.03 Ma) and the murine sample at Bassi represents *A. chinjiensis* as well (Chron C5Ar.1n, ~ 12.73–12.77 Ma). The Dehari locality, situated above S15, lies within the stratigraphic interval encompassing reversal R4, Chron C5Ar.1r, ~ 12.47–12.73 Ma. The biochronological age of Dehari is estimated between 12.4 and 11.6 Ma, which is slightly younger than Chron C5Ar.1r, particularly given Dehari's stratigraphic placement near the middle of the reversal.

**Option 2.** Within this framework, R1 is correlated with C5Ar.3r (12.887 to 13.032 Ma), R2 with C5Ar.2r (12.770 to 12.829 Ma), N4 with C5An.2n (12.272 to 12.474 Ma), and R4 with C5An.1r (12.174 to 12.272 Ma). This correlation places Rashole within polarity zone R1 (C5Ar.3r, ~ 12.887 to 13.032 Ma), which aligns well with its biostratigraphic age estimate of 13.1 to 12.7 Ma based on *P. indicus*. This correlation also places Sunetar 2 within C5Ar.3r, indicating that the primate specimens *Sivapithecus*, *Kapi*, and *Ramadapis* from Rashole and Sunetar 2 date to an age between 12.887 and 13.032 Ma. This age estimate is slightly older than the currently accepted age for the earliest *Sivapithecus* specimens

from the Potwar Plateau, which fall within C5Ar.1n (~12.73 to 12.77 Ma) [22,24,48–50], potentially extending the FAD of *Sivapithecus* by ~100–200 kyr.

Both Tarmin Peak (above S12) and Bassi (above S14) again correlate with chrons R2 and N4, respectively, consistent with biochronological estimates if both *Antemus chinjiensis* and *Megacricetodon* are present at Tarmin Peak (Chron C5Ar.2r, ~12.77–12.82 Ma) and the murine sample at Bassi (Chron C5An.2n, ~12.27–12.47 Ma) represents either late occurring *Antemus chinjiensis* (filling in the previously noted gap between ~12.4–12.7 Ma) or post-*Antemus*. In this correlation, the Dehari locality, situated above S15 and exhibiting a biostratigraphic age of 12.4 to 11.6 Ma, fits well within the age range of chron R4 (12.174 to 12.272 Ma).

**Option 3.** This correlation aligns R1 with chron C5Ar.2r (12.770–12.829 Ma), R2 with C5Ar.1r (12.474–12.735 Ma), N4 with C5An.1n (12.049–12.174 Ma), and R4 with C5r.3r (11.657–12.049 Ma). In this framework, the Rashole biochronological age estimate (12.7–13.1) corresponds well with the age of R1 (~12.770–12.829 Ma), and the first appearance datum (FAD) of primates from Rashole (*Sivapithecus*) and Sunetar 2 (*Ramadapis*, *Kapi*) would fall within this chron, up to 100 kya older than the recorded FAD for *Sivapithecus* on the Potwar Plateau (12.73–12.77 Ma) [22,23,48–50].

In this option, the Tarmin Peak biochronological age estimate (12.7–13.8Ma) based on *A. chinjiensis* and *Megacricetodon* aligns with R2, as the locality lies in the lower part of R2 (~12.7 Ma in this correlation). This scenario places Bassi within chron N4 (12.049–12.174 Ma), which is consistent if the identified *A. chinjiensis* specimens at Bassi instead represent post-*Antemus*. The Dehari locality, with a biochronological age estimate of 11.6–12.4, aligns well with chron R4 (11.657–12.049 Ma).

Within the local magnetic polarity stratigraphy, a reversed-to-normal transition is recorded at the top levels of P18 (321 m) and a normal-to-reversed transition is recorded at the top levels of S9 (156 m). Using each possible correlation option to the GPTS, we determined the sediment accumulation rates for the Ramnagar local magnetic polarity stratigraphy and then extrapolated the sediment accumulation rates (SAR) through the entire section (Fig 9, S5 Table, see S11 Fig. for SAR of option 1and 3). In the three possible scenarios listed above, this 165 m thick sequence would be deposited between 12.474–13.636 (18.55 cm/kyr, Option 1), 12.174–13.032 Ma (19.23 cm/kyr; Option 2), 11.657–12.829 Ma (14.07 cm/kyr; Option 3). The Chinji lithofacies in the Khaur and Tatrot-Andar sections on the Potwar plateau exhibit mean sediment accumulation rates (SAR) of approximately 23 cm/kyr and 13 cm/kyr, respectively [21]. All three of our estimated SAR values for Ramnagar fall within this range, supporting the consistency of sedimentation rates across comparable lithostratigraphic units at Ramnagar and Potwar. Using the SAR rates, we extrapolated the age of the top and bottom of the section compared it to two additional chronologically constrained localities. The lithology at Dalsar (above S22, ~440 m level) was assigned to the Middle Siwaliks sequence by Basu [7] and confirmed in this study as lithologicaly most consistent with the Middle Siwaliks (S2-S3 Fig.). The S22 sandstone (immediately below Dalsar) is the only pebbly sandstone in the Ramnagar sequence described here and it fits best with the lithological description of the Late Miocene Middle Siwalik sequence. Barring any unrecognized disconformity, the lithology below S19 is most similar to the Middle Miocene Lower Siwalik sequence, which would place S19-22 at/near the base of the Middle Siwalik sequence. The base of the Middle Siwaliks sequence coincides with the start of the Late Miocene at ~11.5 Ma on the Potwar Plateau [24]. Thus, the site of Dalsar is estimated to be close to this age as well.

The biostratigraphy of rodents and the macrofauna from the base of the Ramnagar Lower Siwaliks sequence at Kulwanta (above S1, ~5 m level) have been used to estimate the site to be ~13.8–13.5 Ma [10,11,18] i.e., equivalent to the base of the Chinji Fm. on the Potwar Plateau, ~14 Ma. Extrapolations of SAR using Option 2 (base of paleomagnetic sequence is C5Ar.3r) provide an age estimate of ~11.55 Ma for Dalsar (it can vary as lithology changes above S19) and ~13.81 Ma for Kulwanta (S5 Table), which is consistent with existing age estimates for these sites. Extrapolations based on the two other scenarios (Option 1 and Option 3) generate estimates for Dalsar of ~11.83 Ma or ~10.82 Ma, and for Kulwanta of ~14.17 Ma or ~14 Ma, respectively. While both estimates for Kulwanta are reasonably close to the known age

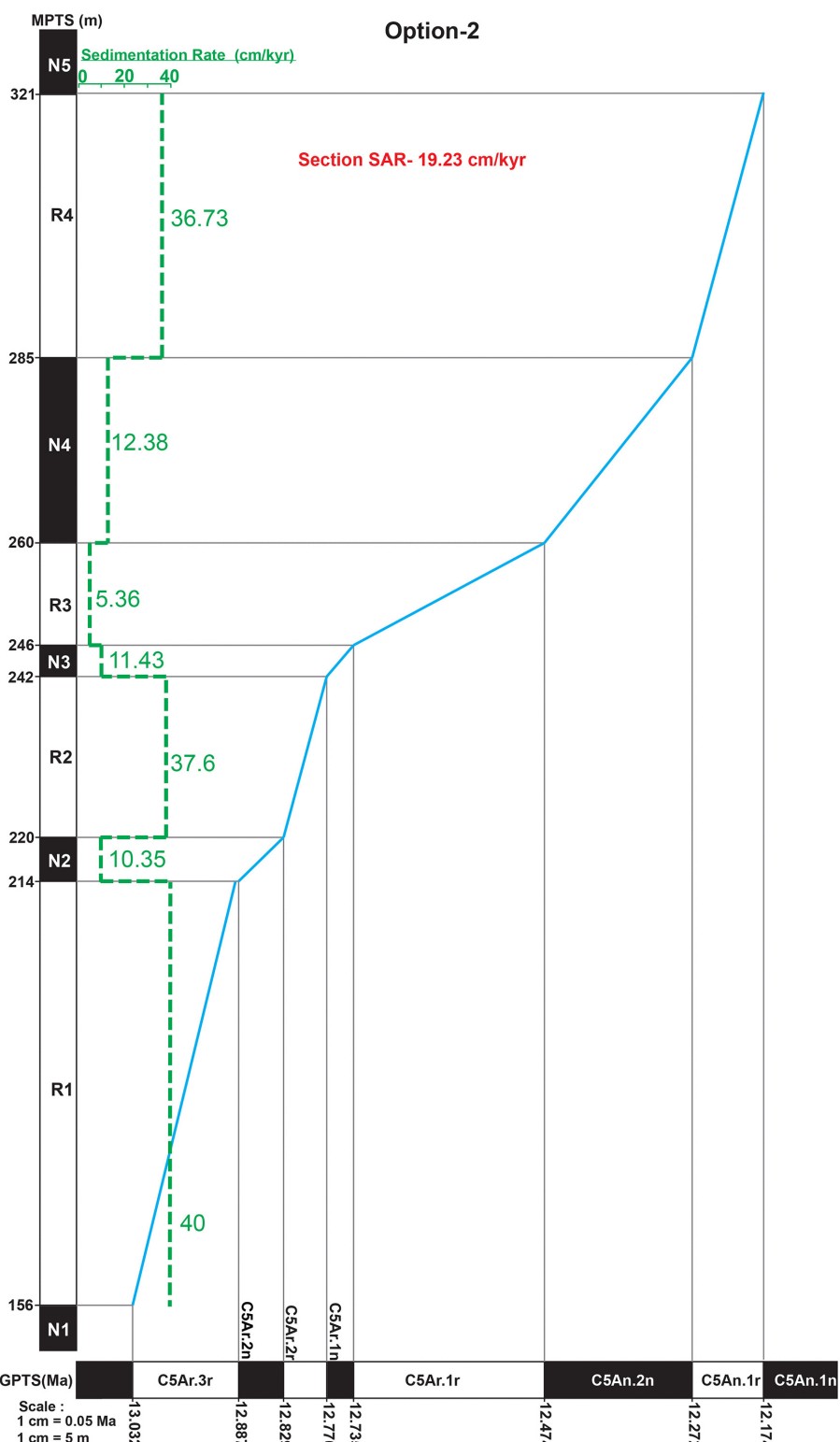

**Fig 9. Observed magnetic polarity stratigraphy (MPTS) is plotted against the standard geomagnetic polarity timescale (GPTS; Ogg, 2020) to assess sediment accumulation rates (SAR) for preferred Option 2.** (For option 1 & 3, see S2 Fig).

of the base of the Chinji Fm., the estimates for Dalsar, which appears to lie at the base of the Middle Siwaliks (~11.5 Ma) on lithological evidence, are either too old (Option 1) or too young (Option 3). Thus, based on the entirety of lithological and biochronological data available, we propose that the most likely paleomagnetic correlation at this time is Option 2. Option 2 is also considered most likely correlation because, as discussed above, it does not require any major changes in the FAD or LAD of established biostratigraphically informative taxa, but rather a slight extension of the rare fossil ape Sivapithecus *indicus*. We consider Option 1, which would be the oldest correlation, to be the most unlikely because it is least consistent with the biostratigraphic and lithostratigraphic data at present. Although, we consider Options 1 and 3 less likely that option 2, we present them as possible alternative age models pending additional paleomagnetic data from above and below the currently sampled sequence.Thus, the 190 m paleomagnetic section that covers the majority of the Ramnagar fossil sites, and all of the Lower Siwalik primate sites, is interpreted here to to GPTS Chrons C5Ar.3r through C5An.1n (>13.032 Ma to <12.174 Ma). This correlation implies that the entire Ramnagar section from Kulwanta to Dalsar covers a time period of ~13.8–11.5 Ma, or the equivalent of the Chinji Formation and the bottom of the Nagri Formation on the Potwar Plateau.

## Implications for the timing of primate evolution

Our preferred paleomagnetic correlation (Option 2) places the FAD of *Sivapithecus indicus* at Rashole within C5Ar3r, between 12.887 and 13.032 Ma, which is~100,000–200,000 years older than the FAD reported from the Chinji Formation in Pakistan at ~12.8 Ma [22,24,28,48–50]. However, any of the three possible correlations options enumerated above results in an earlier FAD for *Sivapithecus*, ranging from ~100 (Option 3) to 500 (Option 1) thousand years earlier than recorded on the Potwar Plateau. An older FAD for *Sivapithecus* at Ramnagar compared to the Potwar Plateau was previously suggested by Sehgal and Patnaik [9], and the biostratigraphic, paleomagnetic, and sedimentological data presented here support their chronological inference, at least in part (i.e., the main section S9-S15). While the current evidence therefore suggests a slight extension to the known *Sivapithecus* time range, it is notable because previous studies have suggested that C5Ar.1n (site Y750 on the Potwar Plateau) likely represents a reliable FAD for *Sivapithecus* in the Lower Siwaliks based on concentrated efforts to find older specimens within the Chinji Formation without success [22,49]. Thus, given the rarity of fossil apes in the Siwaliks, and the relatively complete sequence and fossiliferous nature of the Potwar Plateau, it has been reasonably hypothesized that the absence of *Sivapithecus* below C5Ar.1n (12.77 Ma) is indicative of a true absence before this time period [22]. While it is unclear why *Sivapithecus* should appear slightly earlier in the Ramnagar sequence relative to the Potwar Plateau, slight taphonomic differences and/or the nature of chance/random sampling in the fossil record are possible explanations. If any of our currently hypothesized correlations are correct, is also possible that *Sivapithecus* specimens may yet be recovered from the lower sections of the Chinji Fm. on the Potwar Plateau.

In addition to *Sivapithecus*, the known age ranges of the stem hylobatid *Kapi ramnagarensis* and the sivaladapid *Ramadapis sahnii,* both found at the site of Sunetar 2, can also be refined. Because both Rashole and Sunetar 2 are derived from the same stratigraphic level just above S9, Sunetar 2 would also correlate to ~12.887–13.032 Ma (Option 2, our preferred correlation). Incorporating the next likely correlation (Option 3), we can slightly expand this estimated time range as anywhere between ~12.770–13.032 Ma. As noted by Gilbert [17], the co-occurrence of *Sivapithecus* and *Kapi* at the same stratigraphic level ~12.770–13.032 Ma has important biogeographic implications for the dispersal of these fossil apes from Africa to Asia shortly after the Miocene Climate Optimum, suggesting that both the greater and lesser ape lineages were dispersing into Asia by this time, perhaps as a part of the same faunal dispersal event.

## Conclusions

This study helps to refine the geochronology of the Ramnagar section and the important primates within the succession. The currently available paleomagnetic, biochronological, and lithostratigraphic data combined with estimated sedimentation rates suggests that Ramnagar likely correlates to a time period equivalent to the entire Chinji Formation and lowest

part of the Nagri Formation on the Potwar Plateau, ~14–11.5 Ma. It constrains almost all the primate yielding sites from Ramnagar between ~13.03 Ma and ~11.55 Ma and, based on the most likely correlation to the GPTS, the FAD of the likely stem hylobatid *Kapi ramnagarensis*, sivaladapid *Ramadapis sahnii*, and great ape *Sivapithecus* can now be placed between ~12.8–13.0 Ma, suggesting the first occurrence of Sivapithecus at Ramnagar occurs 100–200 kyr earlier than on the Potwar Plateau.

## Supporting information

**S1 Fig. Section near Songer River bridge toward Sunetar, located approximately 30 meters above S9.**
(TIF)

**S2 Fig. Pebbly sandstone unit S22, identified as the oldest sandstone of the Middle Siwalik.**
(TIF)

**S3 Fig. Rounded pebble embedded within unit S22.**
(TIF)

**S4 Fig. Measuring paleosol P15 using Jacob staff and Abney level.**
(TIF)

**S5 Fig. Continuity of sandstone unit S1 toward the dip direction, similar to S9 and S15.**
(TIF)

**S6 Fig. Continuity of sandstone unit S1 toward the dip direction and P1 Above S1 near Kulwanta Government School.**
(TIF)

**S7 Fig. Uppermost part of paleosol P4 situated above sandstone unit S4, representing fossil locality Ritti-Mitti-2.**
(TIF)

**S8 Fig. Thaplal primate site (P18) located above sandstone unit S15.**
(TIF)

**S9 Fig. Thick paleosol (P17) situated between S14 and S15, near Bassi locality.**
(TIF)

**S10 Fig. Occlusal view of RRA-123, left dp 4 -m1 of Paraulacodus indicus from Rashole.** Thryonomyid dental terminology and measurements used for RRA-123 lower dentition.
(TIF)

**S11 Fig. Observed magnetic polarity stratigraphy (MPTS) is plotted against the standard geomagnetic polarity timescale (GPTS; Ogg, 2020) to assess sediment accumulation rates (SAR) for Options 1 & 3.**
(TIF)

**S1 Table. Palemagnetic data sampling information.**
(XLSX)

**S2 Table. Least Square data of each sample and VGP.**
(XLSX)

**S3 Table. Site Mean and VGP.**
(XLSX)

**S4 Table. Comparative measurements of Megacricetodon and Democricetodon relative to Dehari cricetid.**
(XLSX)

**S5 Table. Sediment accumulation rate of three possible scenarios and Extrapolated Age.**
(XLSX)

## Author contributions

**Conceptualization:** Deepak Choudhary, Christopher C. Gilbert, Daniel J. Peppe, Rajeev Patnaik.

**Data curation:** Deepak Choudhary, Christopher C. Gilbert, Christopher J. Campisano, Daniel J. Peppe, Rajeev Patnaik.

**Formal analysis:** Deepak Choudhary, Daniel J. Peppe, Mohammad Arif.

**Funding acquisition:** Deepak Choudhary, Christopher C. Gilbert, Biren A. Patel, Rajeev Patnaik.

**Investigation:** Deepak Choudhary.

**Methodology:** Deepak Choudhary, Daniel J. Peppe, Rajeev Patnaik.

**Project administration:** Christopher C. Gilbert.

**Resources:** Christopher C. Gilbert, Christopher J. Campisano, Daniel J. Peppe, Rajeev Patnaik.

**Software:** Daniel J. Peppe.

**Supervision:** Christopher C. Gilbert, Christopher J. Campisano, Daniel J. Peppe, Rajeev Patnaik.

**Validation:** Christopher C. Gilbert, Christopher J. Campisano, Daniel J. Peppe, Mohammad Arif, Binita Phartiyal, Ningthoujam Premjit Singh, Rajeev Patnaik.

**Visualization:** Christopher J. Campisano, Daniel J. Peppe.

**Writing – original draft:** Deepak Choudhary.

**Writing – review & editing:** Christopher C. Gilbert, Christopher J. Campisano, Daniel J. Peppe, Mohammad Arif, Sarvendra Pratap Singh, Kahsay Tesfay, Biren A. Patel, Wasim Abass Wazir, Rohit Kumar, Binita Phartiyal, Ningthoujam Premjit Singh, Kongrailatpam Milankumar Sharma, Rajeev Patnaik.

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
