## [Decision Letter · Decision Letter 0]

7 Jan 2026

Dear Dr. Deepak Choudhary,

Thank you for submitting your manuscript to PLOS ONE. After careful consideration, we feel that it has merit but does not fully meet PLOS ONE’s publication criteria as it currently stands. Therefore, we invite you to submit a revised version of the manuscript that addresses the points raised during the review process.

We look forward to receiving your revised manuscript.

Kind regards,

Shamim Ahmad, PhD

Academic Editor

PLOS One

2. In your manuscript, please provide additional information regarding the specimens used in your study. Ensure that you have reported human remain specimen numbers and complete repository information, including museum name and geographic location.

For more information on PLOS One's requirements for paleontology and archeology research, see https://journals.plos.org/plosone/s/submission-guidelines#loc-paleontology-and-archaeology-research.

“Funding generously provided by: Leakey Foundation Grant S202410509; NSF BCS Awards 1945736, 1945618,1945743; MoES/P.O.(Geosci)/46/2015 and SERB-HRR/2018/000063; PSC-CUNY faculty award program; Hunter College; AAPA professional development grant program; USC; IHO/ASU.”

“•          D.C. received support from the Leakey Foundation (https://leakeyfoundation.org), grant S202410509.

•            C.G., C.C., B.P., and R.P. received support from the National Science Foundation (NSF) (https://www.nsf.gov), awards BCS‑1945736, BCS‑1945618, and BCS‑1945743.

•            R.P. received support from the Ministry of Earth Sciences, Government of India (https://moes.gov.in), grant MoES/P.O.(Geosci)/46/2015, and from the Science and Engineering Research Board (SERB), Government of India (https://www.serb.gov.in), grant HRR/2018/000063.

•            C.G. received support from the PSC‑CUNY Faculty Award Program, Hunter College (https://www.rfcuny.org).

•            B.P. received support from the American Association of Biological Anthropologists (AABA) Professional Development Grant Program (https://physanth.org), the University of Southern California (USC) (https://www.usc.edu), and the Institute of Human Origins, Arizona State University (IHO/ASU) (https://iho.asu.edu).

The sponsors or funders had no role in study design, data collection and analysis, decision to publish, or preparation of the manuscript.”

Additional Editor Comments:

Dear Dr Deepak Choudhary,

This manuscript has been reviewed by three independent reviewers, and the authors are kindly requested to carefully go through the comments provided and address them appropriately. The manuscript requires revision before it can be considered for acceptance.

This study presents an important and timely contribution by establishing, for the first time, a magnetostratigraphic framework for the middle Miocene primate-bearing fossil sites of Ramnagar, India. The work fills a critical gap in Siwalik chronostratigraphy, which has been extensively developed for the Potwar Plateau in Pakistan but has remained poorly constrained for Indian localities despite their rich vertebrate fossil record, including key hominoids such as Sivapithecus, Kapi, and Ramadapis. The integration of paleomagnetic data with biochronological evidence is a major strength of the manuscript, and the proposed age bracket of 12.88–13.03 Ma for the primate-bearing horizons is particularly significant, as it pushes back the known first appearance datum (FAD) of Sivapithecus by approximately 200 kyr.

With respect to the paleontological interpretation, I agree with the authors that the option 2–based correlation appears to be the most plausible and internally consistent with the currently known murine fossil record. However, the biochronological constraint seems to rely heavily on the identification and interpretation of relatively scarce Antemus-like fossil material. If this taxonomic assignment were to be revised or proven incorrect, the robustness of the option 2 correlation could be weakened. In this context, the manuscript would be strengthened by the inclusion of photographs or detailed illustrations of the relevant fossil material, allowing readers to better evaluate the biological evidence supporting this correlation.

Additionally, the discussion presented in lines 510–520 is somewhat difficult to follow. This section would benefit from refinement and clarification, particularly with respect to explicitly explaining why the option 2–based correlation is preferred over alternative correlations and how it best reconciles the magnetostratigraphic and biochronological data.

Finally, the authors note that pedogenic modification is recognized within the paleosol units. Given that many of the Ramnagar fossils are recovered from paleosol horizons, a more detailed pedogenic description would be highly valuable. Expanded discussion of paleosol characteristics (e.g., horizon development, carbonate accumulation, root traces, or other pedogenic features) would provide important context for fossil preservation and enhance the paleoenvironmental interpretation.

With Regards,

Dr S Ahmad

Reviewers' comments:

Reviewer's Responses to Questions

**Comments to the Author**

1. Is the manuscript technically sound, and do the data support the conclusions?

Reviewer #1: Yes

Reviewer #2: Yes

Reviewer #3: Yes

2. Has the statistical analysis been performed appropriately and rigorously?

Reviewer #1: Yes

Reviewer #2: Yes

Reviewer #3: N/A

3. Have the authors made all data underlying the findings in their manuscript fully available?

Reviewer #1: Yes

Reviewer #2: Yes

Reviewer #3: Yes

4. Is the manuscript presented in an intelligible fashion and written in standard English?

Reviewer #1: Yes

Reviewer #2: Yes

Reviewer #3: Yes

Reviewer #1: Dear Editor,

This is an interesting article that provide a magnetostratigraphic framework for primate-bearing fossil sites in the middle Miocene of India. Siwalik deposits are well described and examined chronologically in the Potwar Plateau, Pakistan, whereas paleomagnetic framework has not be installed in the middle Miocene localities of Ramnagar, India, even though important fossil localities, including ones with fossil primates (Sivapithecus, Kapi and Ramadapis), have been reported. This study established magnetostratigraphy of the fossil-yielding sites in Ramnagar and dated the primate sites to be between 12.88 and 13.03 Ma. Their results push back the known FAD of Sivapithecus back by 200,000 years.

Here are my comments relevant to paleontology.

I agree with the authors that option 2-based correlation is most plausible and consistent with the fossil record of murines. My understanding is that the biochronological estimateof the fossil record is dependent on the interpretation of scarce materials of Antemus-like fossils. If this assignment is incorrect, option 2 is not the most likely correlation. Is it correct? In this case, I would appreciate if photos of the fossils are available in this manuscript so that biological evidence is clear. Also, I found the discussion in lines 510-520 a bit difficult to follow. I would appreciate if the authors refine the discussion to decipher why option 2-based correlation is most plausible.

The authors wrote that pedogenic modification is recognized in the paleosol units. I think pedogenic description would be appreciated as most fossils from Ramnagar are found from paleosol levels.

P6, line 127: Isnt “cross-section” more common than “transverse” for a geological map?

P7, line 140: delete a comma after Figs. 2.

P8, Table 1: In this table, primate taxa are associated with key sandstones. However, most fossils are produced from paleosol. I thought this table is misleading and needs to be changed. The label “New Section” must be mistaken.

Figure 4: I just wanna let you know that in the NRM thermal demagnetization graph, open circles are too small to see.

P18, line 357-358: “reversals” to “intervals”

P25, line 517: Insert “the” before most likely

P22, Line 451: I would say “dated to an age between”.

Reviewer #2: First of all, the section (190 m) is very small for Magnetostratigraphic study. Getting several reversals within 190 m section is questionable, however, the authors tried to give various reasons and divided the interpretation in three options, but I must warn the authors that this is not a good practice to adjust the local MPS correlation with GPTS in this fashion using Bio (fossils) rather I would suggest to use reversal polarity events for the correlation with GPTS. The sampling part is not sufficient enough to get an average mean VGP latitude. I have made several comments in the annotated pdf manuscript.

Reviewer #3: The authors have utilised multiple evidences on lithostratigraphy, magnetostratigraphy and biochronology to provide an age context to the Lower Siwalik deposits at Ramnagar (north India) and its correlation to the Chinji and Nagri formations (Potwar Plateau) in a GPTS framework. An age range between ~13.03 Ma and ~11.59 Ma for the various primate yielding localities of Ramnagar (north India) has been constrained in the study. An integrated research on stratigraphy by the authors holds significance, as it may assist in better understanding the Neogene primate evolution of the subcontinent in a palaeobiogeographic context. All figures are of good quality. In view of the above, the manuscript should be accepted for publication.

I do have a few minor suggestions for the authors, and believe that they can be addressed during the proofing stage.

Minor suggestions:

Maintain consistency (as per Fig 7, and up to 2 decimal places) throughout the text while mentioning the numerical ages and/or age ranges. For instance, the FAD of the likely stem hylobatid Kapi ramnagarensis, sivaladapid Ramadapis sahnii, and the great ape Sivapithecus: 12.88-13.03 Ma and/or 12.8-13.0 Ma (refer to sections ‘Abstract’, ‘Implications for the timing of primate evolution’, ‘Conclusions’).

Vivesh Vir Kapur

**Do you want your identity to be public for this peer review?** For information about this choice, including consent withdrawal, please see our Privacy Policy

Reviewer #1: No

Reviewer #2: No

Reviewer #3: No

---

## [Author Response · Author response to Decision Letter 1]

4 Feb 2026

Date: January 31, 2026

Journal: PLOS ONE

Manuscript ID: PONE-D-25-57067

Title: Geochronological Insights of Middle Miocene Primates and Vertebrate Fauna of Ramnagar (J&K, India): Integrating Litho- and Magnetostratigraphy.

To: The Academic Editor and Reviewers

Subject: Response to Reviewers

Dear Editor and Reviewers,

We wish to express our sincere gratitude to the reviewers for their thorough and constructive evaluation of our manuscript. We have carefully considered each comment and have revised the manuscript accordingly. We believe these changes have significantly strengthened the paper.

As requested, we have provided:

1. Response to Reviewers (this document)

2. Revised Manuscript with Track Changes (highlighting all modifications)

3. Manuscript (a clean version with all changes accepted)

Below is our point-by-point response to the reviewer comments.

Part 1: Response to Reviewer #1

Reviewer Comment 1: I agree with the authors that Option 2-based correlation is most plausible... My understanding is that the biochronological estimate... is dependent on the interpretation of scarce Antemus-like fossils... If this assignment is incorrect, option 2 is not the most likely correlation. Is it correct? In this case, I would appreciate if photos of the fossils are available... Also, I found the discussion in lines 510-520 a bit difficult to follow.

Author Response: We thank the reviewer for their agreement that Option 2 is the most plausible correlation. We would like to clarify the points raised regarding the fossil evidence and discussion:

1. Dependence on Antemus-like fossils: We clarify that our biochronological assessment and preference for Option 2 is not dependent on the presence or absence of Antemus, and instead incorporates numerous time-constrained rodent taxa. Regarding Antemus more specifically, it is true that some options are more likely than others based on the ultimate identification of some small samples from various localities as Antemus or post-Antemus, and again these are enumerated in the Discussion section of the manuscript. Regarding Option 2, the possibilities that the small sample of Bassi specimens represents Antemus OR “post-Antemus”, a form more derived than Antemus chinjiensis, are discussed, and given the known rodent fossil time-gap between the LAD of A. chijiensis and post-Antemus (~12.7-12.4 Ma), either taxonomic possibility is consistent with the age proposed in Option 2, whereas Option 1 is most consistent with Antemus chinjiensis and Option 3 is most consistent with post-Antemus. Therefore, the plausibility of Option 2 is not negatively affected by the taxonomy of these specimens, whether ultimately determined to be Antemus or post-Antemus. We discuss these scenarios more fully in the text.

2. Request for Fossil Photos:

o Regarding the Antemus specimens: As these specimens have been previously described and illustrated in published literature, we have provided the necessary citations (Lines 395–406) rather than reproducing the images in the main text.

o Regarding Paraulacodus indicus: At the time of our initial submission, the descriptive paper for this species [27] was under review. It has since been published. However, to ensure the biological evidence is immediately clear to readers as requested, we have retained the photograph of P. indicus in Supplementary Figure 10.

3. Discussion Clarity (Lines 510–520): We have revised the discussion section to improve readability. Specifically, we have rearranged the paragraph order to create a more logical flow, making the argument for Option 2 easier to follow.

Reviewer Comment 2: P6, line 127: Isnt “cross-section” more common than “traverse” for a geological map?

Author Response: We appreciate the reviewer’s suggestion. However, we have retained the term “traverse” because it specifically describes the linear path followed during fieldwork to measure the stratigraphic thickness. This is distinct from a “cross-section,” which typically implies a constructed interpretation of the subsurface geometry. In this context, "traverse" more accurately reflects the methodology used to collect the data.

Reviewer Comment 3: P7, line 140: delete a comma after Figs. 2.

Author Response: Corrected as suggested.

Reviewer Comment 4: P8, Table 1: In this table, primate taxa are associated with key sandstones. However, most fossils are produced from paleosol. I thought this table is misleading and needs to be changed. The label “New Section” must be mistaken.

Author Response: We thank the reviewer for highlighting this ambiguity. We have revised Table 1 to explicitly separate the lithological markers (sandstones) from the fossiliferous strata (paleosols). Specifically, we added a new column titled "Fossiliferous Paleosol Horizon" to link primate taxa directly to their specific paleosol layers rather than the sandstone units. We also corrected the header to "New Section (This Study)" and updated the caption to reflect these distinctions.

Reviewer Comment 5: Figure 4: I just wanna let you know that in the NRM thermal demagnetization graph, open circles are too small to see.

Author Response: We appreciate the reviewer’s observation. We have double-checked the original high-resolution (300 DPI) TIFF file submitted for publication, and the open circles are clearly distinguishable. The visibility issue may be due to the compression of the PDF generated for the review process. We have chosen to retain the current symbol size to ensure that closely spaced data points remain distinct and do not obscure one another.

Reviewer Comment 6: P18, line 357-358; The reviewer suggested changing "reversals" to "interval" in the sentence:

Author Response: We agree with the reviewer. The term "polarity intervals" is stratigraphically more accurate than "reversals" when referring to the magnetozones defined in the section. We have amended the text accordingly in the revised manuscript

Reviewer Comment 7: P22, Line 451: I would say “dated to an age between”

Author Response: Corrected as suggested.

Reviewer Comment 8: P25, line 517: Insert “the” before most likely

Author Response: Corrected as suggested.

Response to Reviewer #2 (General Comment)

Reviewer Comment: First of all, the section (190 m) is very small all for Magnetostratigraphic study. Getting several reversals within 190 m section is questionable... I must warn the authors that this is not a good practice to adjust the local MPS correlation with GPTS in this fashion using Bio (fossils) rather I would suggest to use reversal polarity events for the correlation with GPTS. The sampling part is not sufficient enough to get an average mean VGP latitude.

Author Response: We thank the reviewer for their critical assessment. We understand the concerns regarding the section thickness and correlation strategy. However, we would like to provide the following justifications for our methodology, which aligns with established practices in terrestrial magnetostratigraphy:

1. Section Thickness (190 m) and Polarity Frequency:

We acknowledge that 190 m is a relatively short section compared to basin-wide studies; however, it accurately represents the specific stratigraphic context of the fossiliferous localities at Ramnagar. The observation of multiple reversals within this thickness is consistent with both the sediment accumulation rate and the known magnetic polarity history of the time period. Specifically, the Middle Miocene is characterized by a high frequency of geomagnetic reversals (Ogg, 2020). Consequently, even a condensed section with a lower sediment accumulation rate would naturally record a higher density of reversal events per meter. We have reported the polarity pattern exactly as preserved in the rock record, and our high-resolution sampling ensures these are accurately identified geomagnetic events rather than artifacts.

2. Correlation Strategy (Integration of Biochronology and GPTS):

We respectfully disagree with the reviewer regarding the utility of biostratigraphic constraints. In terrestrial sequences like the Siwaliks, the magnetic polarity pattern is often non-unique and effectively acts as a "bar code" that repeats. Without independent anchor points, such as isotopic ages or biostratigraphic markers, it is impossible to uniquely correlate a short magnetic section to the GPTS.

Our methodology follows the established chronostratigraphic protocols developed for the Siwalik Group. Notably, in the adjacent Potwar Plateau of Pakistan, a robust and widely accepted geochronological framework has been constructed precisely by integrating magnetostratigraphy with a well-established rodent biochronology [21-26]. We have applied this same proven regional methodology to the Ramnagar section.

Therefore, using faunal constraints to filter valid correlation options is not "adjusting" the data but is rather the standard method of integrated magneto stratigraphy. We presented three options to ensure scientific transparency, demonstrating that Option 2 is currently the most likely scenario the only scenario that satisfies both the objective magnetic polarity pattern and the independent biological evidence.

3. Sampling Sufficiency:

We maintain that our sampling density is sufficient to determine reliable magnetic polarity stratigraphy. Although the section is stratigraphically short, the site mean directions yield robust statistical parameters (i.e, Fisher precision parameter k and α95 indicating coherent clustering of magnetic vectors).

Furthermore, the primary objective of this study was to establish a polarity-based chronological framework for the specific fossil horizons, rather than to conduct a detailed study of geomagnetic secular variation. Given the statistical rigor of the site mean directions, the derived VGP latitudes are sufficiently robust to distinguish clearly between Normal and Reverse polarity states, which is the requisite standard for magnetostratigraphic correlation

Part 2: Response to Reviewer #2 (PDF Annotations)

Reviewer Comment: Refers to the annotated PDF manuscript.

Author Response: We have carefully reviewed all the annotations provided in the PDF. We have incorporated the majority of the suggested editorial and grammatical corrections into the revised manuscript. In a few specific instances, we have chosen to retain the original phrasing to ensure the precise scientific meaning is conveyed or to maintain consistency with our terminology. All substantive scientific comments raised in the annotations have been addressed in the point-by-point response above.

Reviewer Comment (L-142): Why the authors have not sampled all 440 m sequence?

Author Response: Already answered in previous comment. Further, at this time, additional sampling and analysis are beyond the scope of the work presented here and would not change the overall conclusions.

Reviewer Comment (Line -145): Does this primate-bearing locations are good enough to ample for paleomagnetic study as there is an interruption in the remanence?

Author Response: We wish to clarify the lithological context of the fossil sites. The primate-bearing horizons in the Ramnagar section predominantly occur as thin layers of pseudoconglomerate intercalated within the mudstone/paleosol units.

For the paleomagnetic study, we specifically targeted the fine-grained matrix and the immediately associated mudstones/paleosols (typically 1 to 40 cm below the sandstone). These fine-grained lithologies provided high-quality data with stable Characteristic Remanent Magnetization (ChRM) directions, effectively ruling out significant secondary overprinting or interruption of the remanence.

Reviewer Comment (line-161): Is this method of lithostratigraphic correlation between fossil localities valid?

Author Response: We are confident that our lithostratigraphic correlations are robust. Importantly, we did not rely solely on paleomagnetism to correlate the localities. Instead, our primary method was physical lithostratigraphic correlation based on extensive fieldwork and walk-over surveys. We validated the continuity of the fossiliferous horizons by conducting three distinct strike-parallel traverses (Thaplal to Dehari, Ramnagar-Dalsar Bridge to Dehari, and Suntar to Dehari). By physically tracing the key marker sandstone beds across these traverses, we established a direct structural connection between the main section and the outlying fossil localities. The paleomagnetic results served as a secondary verification, and confirm the stratigraphic equivalence established by our field

Reviewer Comment (L-163): Were the samples consolidated or friable? Needs an explanation as the authors mentioned that they collected block samples and sandstones/mudstones are fine grained and mostly are unconsolidated state.

Author Response: We respectfully refer the reviewer to Lines 176–178 of the manuscript, where this explanation is already provided

Reviewer Comment (L-165-167): I don't think 2-3 samples per site is good enough for Magnetostratigraphy study. I prefer minimum of 5 samples per site would give good average of the site VGP latitude.

Author Response: We appreciate the reviewer’s suggestion regarding sample density. While we agree that higher sample counts are advantageous, our sampling strategy aligns with established methodologies for high-resolution terrestrial magnetostratigraphy (see for example Zeigler and Kodama, 2017, https://doi.org/10.1016/B978-0-12-803243-5.00005-4; Tauxe et al., 2018, https://earthref.org/MagIC/books/Tauxe/Essentials/. ). Importantly, recent studies in comparable terrestrial successions (e.g., Flynn et al., 2020, ( https://doi.org/10.1130/B35481.1.) have successfully established robust age models using site means derived from a minimum of three reliable samples.

Following the protocol of Flynn et al. (2020), we prioritized statistical reliability over raw quantity. We validated our site means using Fisher statistics, accepting only those sites where the clustering of vectors demonstrated a clear characteristic remanence (ChRM) distinct from random scatter. Sites that failed to meet these rigorous statistical criteria (high α95 or low k) were excluded from the analysis.

Therefore, even with N=3, the statistical coherence of our site means combined with their stratigraphic consistency confirms that they accurately record the primary geomagnetic polarity.

Reviewer Comment: Why Alternating Field demagnetization (AFD) was not carried out?

Author Response: Alternating Field (AF) demagnetization was not utilized because the primary magnetic carrier in these oxidized sediments is high-coercivity hematite, which typically resists standard AF peak fields (>100 mT). We therefore employed stepwise thermal demagnetization, the established protocol for red beds, which successfully isolated the stable Characteristic Remanent Magnetization (ChRM) up to 680°C.

Reviewer Comment (L-198-199): Using which Instrument?

Author Response: The details of the instrumentation are already provided in the "Methodology" section of the manuscript. As specified in Lines 222–223, all thermal demagnetization procedures were conducted using an ASC Scientific controlled atmosphere thermal demagnetizer in an inert nitrogen (N2) atmosphere to prevent oxidation

Reviewer Comment (L-205-206): The ChRM was extracted from at least four remanent vector endpoints aligned linearly toward the origin and had a maximum angle of deviation (MAD) < 20°. (Reviewer: it too high)

Author Response: We appreciate the reviewer’s attention to our data selection criteria. We wish to clarify that while we set a conservative upper threshold of MAD < 20°, the vast majority of our specimens yielded significantly higher precision. Specifically, 96.7% of our analyzed samples possess a MAD < 15°, with only a minor fraction (3.3%) falling between 15° and 20°. We retained the < 20° cutoff to remain consistent with established methodologies for terrestrial magnetostratigraphy in similar depositional settings. For example, Flynn et al. (2020) employed a MAD < 20° threshold for Paleocene terrestrial sediments in the San Juan Basin. Furthermore, our criteria were stricter than the cited study regarding linearity; whereas Flynn et al. (2020) accepted lines based on three points, we required a

---

## [Editor Report · Decision Letter 1]

8 Feb 2026

Geochronological Insights of Middle Miocene Primates and Vertebrate Fauna of Ramnagar (J&K, India): Integrating Litho- and Magnetostratigraphy

PONE-D-25-57067R1

Dear Dr. Choudhary,

We’re pleased to inform you that your manuscript has been judged scientifically suitable for publication and will be formally accepted for publication once it meets all outstanding technical requirements.

Kind regards,

Shamim Ahmad, PhD

Academic Editor

PLOS One

Additional Editor Comments (optional):

Dear Dr Choudhary,

We are pleased to inform you that your manuscript, “Geochronological Insights of Middle Miocene Primates and Vertebrate Fauna of Ramnagar (J&K, India): Integrating Litho- and Magnetostratigraphy,” has been accepted for publication. Congratulations, and thank you for choosing PLOS ONE. We wish you continued success in your future research endeavors.

Sincerely,

Dr S Ahmad
---

## [Editor Report · Acceptance letter]

PONE-D-25-57067R1

PLOS One

Dear Dr. Choudhary,

I'm pleased to inform you that your manuscript has been deemed suitable for publication in PLOS One. Congratulations! Your manuscript is now being handed over to our production team.

Kind regards,

on behalf of

Dr. Shamim Ahmad

Academic Editor

PLOS One